# Performance analysis of regional AquaCrop (v6.1) biomass and surface soil moisture simulations using satellite and in situ observations

Shannon de Roos[1], Gabriëlle J. M. De Lannoy[1], Dirk Raes[1]

[1]Department of Earth and Environmental Sciences, KU Leuven, Heverlee, B-3001, Belgium

*Correspondence to*: Shannon de Roos (Shannon.deroos@kuleuven.be)

**Abstract.** The current intensive use of agricultural land is affecting the land quality and contributes to climate change. Feeding the world's growing population under changing climatic conditions demands a global transition to more sustainable agricultural systems. This requires efficient models and data to monitor land cultivation practices at the field to global scale. This study outlines a spatially distributed version of the field-scale crop model AquaCrop version 6.1, to simulate agricultural biomass production and soil moisture variability over Europe at a relatively fine resolution of 30 arcseconds (~1 km). A highly efficient parallel processing system is implemented to run the model regionally with global meteorological input data from the Modern-Era Retrospective analysis for Research and Applications, version 2 (MERRA-2), soil textural information from the Harmonized World Soil Database, version 1.2 (HWSDv1.2), and generic crop information. The setup with a generic crop is chosen as a baseline for a future satellite-based data assimilation system. The relative temporal variability in daily crop biomass production is evaluated with the Copernicus Global Land Service dry matter productivity (CGLS-DMP) data. Surface soil moisture is compared against NASA Soil Moisture Active Passive surface soil moisture (SMAP-SSM) retrievals, the Copernicus Global Land Service surface soil moisture (CGLS-SSM) product derived from Sentinel-1, and in situ data from the International Soil Moisture Network (ISMN). Over central Europe, the regional AquaCrop model is able to capture the temporal variability in both biomass production and soil moisture, with a spatial mean temporal correlation of 0.8 (CGLS-DMP), 0.74 (SMAP-SSM) and 0.52 (CGLS-SSM), respectively. The higher performance when evaluating with SMAP-SSM compared to Sentinel-1 CGLS-SSM is largely due to the lower quality of CGLS-SSM satellite retrievals under growing vegetation. The regional model further captures the short-term and interannual variability, with a mean anomaly correlation of 0.46 for daily biomass, and mean anomaly correlations of 0.65 (SMAP-SSM) and 0.50 (CGLS-SSM) for soil moisture. It is shown that soil textural characteristics and irrigated areas influence the model performance. Overall, the regional AquaCrop model adequately simulates crop production and soil moisture and provides a suitable setup for subsequent satellite-based data assimilation.

## 1 Introduction

Over the past 60 years, global agricultural production has more than tripled (FAO, 2017). This became possible through productivity-enhanced technologies, industrialization and expansion of agricultural land. However, the current intensive use of cropland is resulting in reduced land quality and increased greenhouse gas emissions, which in turn impact agricultural systems (Foley et al., 2011; Kopittke et al., 2019). To meet the future crop demand of a vastly growing population, while minimizing the ecological footprint and increasing the crop resilience for changing climatic conditions, the need to adapt to more effective and sustainable land cultivation practices is urgent (Aznar-Sánchez et al., 2019; Pingali, 2012; Raes and Vanuytrecht, 2017).

To evaluate the effect of environmental conditions and different management practices on crop production, there exists a variety of models that simulate the biophysiological growth of crops at the field scale. An overview of 70 of such crop models is given by Di Paola et al. (2016). Some of these point-based crop models have more recently been upscaled and assessed at a regional to global level (Balkovic et al., 2013, Boogaard et al., 2013, Folberth et al. 2019, Liu et al., 2007, Müller et al., 2017, Nichols et al., 2011, Resop et al., 2012, Roerink et al., 2012, Stöckle et al., 2014). Large-scale crop models are a valuable asset in providing information to policy makers and for applications in climate scenario analyses (Asseng et al., 2013, Iizumi et al., 2018). A downside of large-scale crop models, especially at a global level, is that they often suffer from the generalization of input data and loss of information that is typically available at smaller scales, resulting in larger errors at the local scale. The AgMIP Global Gridded Model Intercomparisons (GGCMI) is a framework initiated to overcome this issue. It is built on a large group of crop modelling researchers that combine and intercompare a set of upscaled point models or global gridded crop models to assess and reduce the bias and uncertainties at a global level (Elliot et al., 2015). Another possibility to improve the simulations at either the local or larger scale, is the updating of the model simulations with remote sensing observations via data assimilation. There are several studies that have already used data assimilation in regional crop modelling systems (De wit & van Diepen, 2007; Mladenova et al., 2019; Zhuo et al., 2019), either for parameter or state updating. Parameter updating or calibration allows to match the absolute values of the simulations with (most often historical) observations. State updating allows to correct the relative temporal evolution and to obtain better initial conditions for subsequent model predictions. To get the most optimal results with data assimilation, it is important to start with a reliable model that is able capture the seasonal as well as interannual temporal variabilities.

This study presents a methodology to apply the original field-scale AquaCrop model version 6.1 efficiently over a large region and for any spatial resolution. The flexible model setup will allow for many different applications, but in this study the focus is on the preparation of a satellite-based data assimilation system. AquaCrop was developed by the FAO to estimate responses of herbaceous crops to water (Raes et al., 2009; Steduto, et al., 2009). It differs from most other crop models by its low requirement of detailed input data, as it aims for a balance between simplicity, accuracy and robustness (Steduto, et al., 2009). The model has been applied in numerous studies for various crop types and environmental conditions and shows good results in simulating crop biomass and yield, especially when calibrated for local field conditions (Abedinpour et al., 2012; Geerts et

al., 2009; Hsiao et al., 2009; Maniruzzaman et al., 2015; Razzaghi et al., 2017; Sandhu and Irmak, 2019). Earlier spatially distributed versions of AquaCrop were developed by e.g. Lorite et al. (2013), Sallah et al. (2019) and Huang et al. (2019), using a Geographic Information System or batch processing with remote sensing data input. Some challenges of existing distributed AquaCrop systems are related to the limited scalability and high computational cost when they are applied to any large domain at any resolution, the limitations in the upscaling of crop parameters from the plant or field to the grid scale (Han et al., 2020), or the availability of other suitable spatially distributed parameters or input information. Applications of the AquaCrop model at a continental scale exist, but are very limited (Dale et al., 2017) and so far are only used in combination with coarse spatial resolutions. To our best knowledge, no study has yet reported on high-resolution and large-scale (beyond country level) applications of AquaCrop.

The continental setup of our regional AquaCrop simulations uses spatially distributed input data about soil texture and meteorology, while assuming a homogenous generic crop. To evaluate, or later update, select variables within such a regional modelling system, in situ data only provide sparse information. However, a range of spatially distributed optical and microwave-based satellite data are available at various temporal and spatial resolutions. A confrontation between model simulations and satellite data to evaluate or update the model simulations is not always trivial. Most importantly, the magnitude of model simulations and satellite retrievals of soil moisture or biomass are often not directly comparable. Biases between models and observations are inevitable, because they represent different quantities (Koster et al., 2009, Reichle et al., 2004) or are simply based on different assumed parameterizations. The assumption of a generic crop will for example lead to inevitable biases. Via parameter estimation, soil and vegetation parameters can be spatially tuned to reduce such biases, but this is often not feasible for satellite retrievals or difficult with more detailed models at the regional to global scale. For this same reason, state-of-the-art data assimilation systems for state updating are designed to correct for random error, and not for systematic bias. Therefore, satellite products of relative soil water indices or anomaly total water storages are often distributed (Wagner et al., 1998, Albergel et al., 2008, De Lannoy et al., 2016, Li et al., 2019), and the performance of large-scale model simulations is often evaluated using bias-free temporal skill metrics (De Lannoy et al., 2015, Gruber et al., 2020).

The objective of this research is to assess whether a high-resolution regional gridded AquaCrop model can capture seasonal, inter-annual and short-term temporal variability, as well as the spatial variability, of biomass and surface soil moisture, when using global spatially distributed input data about soil texture and meteorology and assuming a generic crop. The model performance will be evaluated over Europe at a spatial resolution of 30 arcseconds (1/120°; ~1 km at the equator), using satellite products derived from both optical and microwave sensors and in situ measurements.

The structure of the paper will be as follows: sections 2 and 3 will cover the methodology, with a description of the regional AquaCrop model setup, the evaluation datasets and performance metrics. In section 4 the results will be presented and discussed, followed by a conclusion in section 5.

## 2 The regional gridded AquaCrop model

### 2.1 AquaCrop equations

AquaCrop is a daily crop-water productivity model that translates, on a daily basis, the simulated amount of crop transpiration into a proportional amount of biomass for a single field, which is assumed to be homogeneous (Raes et al., 2009, Steduto et al., 2009). The relation between transpiration and biomass production is defined by a Water Productivity (WP) factor:

$$B = WP^* . \sum \frac{Tr}{ET_o} \tag{1}$$

$B$ (t ha$^{-1}$) is the cumulative biomass produced, $WP^*$ is the WP (g m$^{-2}$) factor normalized for atmospheric $CO_2$ (369.41 ppm for the year 2000) and for climate, and $Tr$ (mm day$^{-1}$) is the transpiration, also normalized for climate after division by the reference evapotranspiration, $ET_0$ (mm day$^{-1}$). Because of this normalization, the $WP^*$ factor only significantly differs between C3 and C4 crops, where C4 crops have a higher WP* due to a more efficient carbon assimilation process. The calculation of $Tr$ is dependent on $ET_0$, the adjusted green canopy cover ($CC^*$; -), the crop transpiration coefficient ($K_{c,tr}$; - ), the cold stress coefficient ($Ks_{tr}$; - ) and the soil water stress coefficient ($Ks$; -).

$$Tr = Ks . Ks_{tr} . K_{c,tr} . CC^* . ET_0 \tag{2}$$

To calculate the soil water balance, AquaCrop divides the soil profile into multiple compartments (default 12) with depth increments Δz (default 0.1 m). For deeper soils, Δz increases exponentially with increasing soil depth, so that the processes of the near surface layers can still be resolved with sufficient detail. The number of compartments is independent of the number of soil horizons and the hydraulic properties for each compartment will be used depending on the soil layer in which they reside. The simulation of the water content in each compartment is done with a set of finite difference equations (subroutines), that are defined in terms of the dependent variable θ, as represented in Eq. 3 (Raes, et al., 2012). First, the drainage of the soil profile is calculated. Then, the water infiltration is computed (after subtraction of surface runoff) and upward movement of water by capillary rise is estimated. Finally, the amount of water lost by evaporation and crop transpiration is subtracted.

$$\theta_{i,j} = \theta_{i,j-1} + \Delta\theta_{DF_{i,\Delta t}} + \Delta\theta_{I_{i,\Delta t}} + \Delta\theta_{CR_{i,\Delta t}} + \Delta\theta_{E_{i,\Delta t}} + \Delta\theta_{T_{i,\Delta t}} \tag{3}$$

where $\theta_{i,j}$ is the soil water content of compartment $i$ at time step $j$, $\theta_{i,j-1}$ is the water content of compartment $i$ at the previous time step and $\Delta\theta_{X_{i,\Delta t}}$ indicate the change in moisture due to various processes $X$, with $X= DF$: downward flow, $I$: infiltration, $CR$: capillary rise, $E$: soil evaporation, $T$: crop transpiration.

Downward flow over the compartments is described by an exponential drainage function (Eq. 4) based on the volumetric water content in the compartment i ($\theta_i$) within the soil layer and drainage characteristics of the soil layer (Raes et al., 2006, Raes et al., 2009):

$$\Delta\theta_{DF_{i,\Delta t}} = \tau_i(\theta_{sat} - \theta_{FC})\frac{e^{\theta_i - \theta_{FC} - 1}}{e^{\theta_{sat} - \theta_{FC} - 1}} \tag{4}$$

$\Delta\theta_{DF_{i,\Delta t}}$ is the decrease in water content over time (m$^3$ m$^{-3}$ d$^{-1}$), $\theta_{FC}$ and $\theta_{sat}$, are the volumetric moisture content at field capacity and at saturation (i.e. the porosity) of the soil layer, and $\tau_i$ is the drainage coefficient derived from the saturated hydraulic conductivity ($K_{sat}$). Infiltration (I) is the sum of water that enters the soil, which is rainfall minus surface runoff, and possibly irrigation. Internal drainage between compartments is defined by the drainage ability, which depends on $\theta_{sat}$ and $\theta_{FC}$ (Eq. 4). The cumulative drainage amount from any compartment will percolate through as long as its drainage ability is greater than

or equal to the drainage ability of the overlying compartment. If the drainage ability is lower than the overlying compartment, the cumulative drainage amount will be stored in that compartment, increasing the water content and thereby its drainage ability. If then the drainage ability is reaching the equal amount of that of the overlying compartment, excess drainage will percolate through to the lower compartment. For the bottom soil compartment, the drainage is lost as deep percolation. The runoff is estimated based on the curve number (CN) method, developed by the US Soil Conservation Service (USDA, 1964).

The CN values are dependent on $K_{sat}$ of the topsoil layer. Upward flow by capillary rise is estimated based on the depth of the groundwater table and hydraulic characteristics of the soil layers. Since no groundwater table is implemented in the regional version of the model in this paper, capillary rise is not included in the simulations. Soil evaporation is based on the soil wetness and crop cover (Ritchie, 1972) and water extraction by roots is described with the sink term from Feddes (1982). Because the root density for most crops is highest near the soil surface and decreases with increasing soil depth, the water extraction pattern

by roots is simulated as follows: 40/30/20/10% for the upper quarter to the lowest quarter of the root zone (Raes et al., 2009). The estimated water retained in the root zone that will be available to the plants (Wr) at each daily timestep is described by the fraction of total available water (TAW) after subtraction of depleted water (Dr). TAW is the difference of volumetric moisture content between field capacity ($\theta_{FC}$) and wilting point ($\theta_{WP}$) over the root zone and is therefore dependent on soil texture and depth.

Plant stresses, such as water excess or water limitation, cold/heat air temperature stress, soil fertility and salinity stresses, can affect biomass production during different steps of the calculation procedure, depending on the process that is affected (i.e. canopy expansion, crop transpiration, pollination). The inclusion of stress factors is done by assigning unique thresholds to each of these biological processes (Raes et al., 2018). Further details on the AquaCrop equations can be found in the calculation procedure manual by Raes et al. (2018).

**2.2 Regional model setup**

The model domain of this study covers the agricultural land in the central part of Europe (35°N-55°N, 10°E-20°E), and 45 pixels across all of mainland Europe where in situ soil moisture data are available for evaluation (three in situ points are also included in the central European domain). The model was run for the years 2011 through 2018, starting on the first of January 2011. The initial soil moisture content for the first year was set at $\theta_{FC}$, since the runs were initiated mid-winter, and for the

subsequent years the initial soil moisture content was based on the moisture content of the last day from the previous year.

Because the evaluation for soil moisture was done with microwave-based satellite products that pertain to the surface layer, the AquaCrop volumetric moisture content of the top soil compartment (WC01), at a depth of 0.05 m (center of top 10 cm) was chosen for evaluation in this study. For the biomass, the daily productivity (t ha$^{-1}$ ) was derived from the cumulative biomass. In the regional version of AquaCrop, a single homogeneous field is represented by a 30 arcsecond (~1-km) pixel, and input and output were defined independently for each pixel. The system can easily be set up for any given resolution over any domain. In this study, the model was run exclusively for dominantly rainfed agricultural areas, based on the land use map of the CORINE Land Cover inventory (Büttner, 2014) for the year 2012. This dataset is available at 100-m resolution and was aggregated to 30 arcseconds. To best represent the pixels as agricultural fields, only pixels were included of which at least 50 CORINE pixels (~50% of one AquaCrop pixel) contained non-irrigated agriculture.

The AquaCrop input data are categorized into several components e.g. climate, soil, vegetation and management. For each component, parameters are described in a text-file with a specific file extension that is recognized by the model. A Project Management (PRM) file oversees all the information for a single field (or pixel) and contains paths and names of these input files. This PRM-file is read and executed by AquaCrop, after which an output file is created.

The original Pascal source code of AquaCrop v6.1 was minimally adjusted and compiled on the Linux-based High-Performance Computer (HPC) of the Vlaams Supercomputer Centrum (VSC), and the resulting executable was plugged into a Python wrapper to allow massively parallel simulations to optimize the model efficiency over larger spatial domains. The current system allows for easy implementations of later versions of AquaCrop. The regional input files have to be prepared before model execution. The Python wrapper then creates the PRM-file for a pixel as a first step of the model run, after which the AquaCrop model is executed and time series output is stored into a new folder for each pixel. The reason for creating the PRM-files right before the model execution is so that changes to a project can be made quickly. With this setup, AquaCrop simulations over 1000 pixels for 8 years can be completed in a wall time of 2.2 minutes when using 36 processors. The runs over the domain and period used in this study were completed in approximately 20 hours on 36 processors.

## 2.3 Model input

The meteorological forcings were extracted from the global Modern-Era Retrospective analysis for Research and Applications, version 2 (MERRA-2; Gelaro et al., 2017). The MERRA-2 meteorological variables have a 3-hourly temporal resolution and a spatial resolution of 0.5° lat x 0.625° lon, and are readily available at a latency of about a month. A nearest neighbour function was used to identify the 30 arcsecond pixels situated within one MERRA-2 grid to assign meteorological input. Minimum and maximum temperature and precipitation were converted into daily data needed for the AquaCrop model. The reference evapotranspiration $ET_0$ was derived from the FAO Penman-Monteith equation, using radiation, wind speed, average temperature and dew temperature from MERRA-2 (Allen et al., 1998). For the FAO Penman-Monteith equation, a psychrometric constant of 0.067 was assumed for the entire domain and variations in topographic elevation were not taken into account. At high elevations (>1 km asl) this could result in deviations of $ET_0$ of max 0.2 mm day$^{-1}$. However, since most agricultural areas are located at much lower elevations, the effect of the psychrometric constant was assumed to be very small.

The long record of mean annual $CO_2$ concentration observed at Mauna Loa (Hawaii, USA) was used as $CO_2$ input (default file in the database of AquaCrop).

The soil texture and organic matter was taken from the Harmonized World Soil Database version 1.2 (HWSDv1.2). The HWSDv1.2 has a spatial resolution of 30 arcseconds. The hydraulic soil properties for 253 different soil classes were linked to the information on mineral soil texture and organic matter from the HWSDv1.2 via pedo-transfer functions as in De Lannoy et al. (2014). More specifically, AquaCrop uses the soil water content at various matrix potentials, i.e. $\theta_{WP}$, $\theta_{FC}$, $\theta_{sat}$, and $K_{sat}$. These parameters are available for a top layer (0-30 cm) and a deeper layer (30-100 cm). Stoniness and soil salinity were not considered. No restrictions on the rootzone development by impermeable layers were included in the simulations. According to the 1-degree global dataset of soil depth to bedrock used by the Second Global Soil Wetness Project (Dirmeyer and Oki, 2002; Mahanama et al., 2015) and the 250 m resolution map developed by Shangguan et al. (2017), the bedrock is generally deeper than 1 m over the study area, which allows for reaching the maximum effective rooting depth.

A soil fertility stress parameter in the field management file provides an indication of the overall soil quality. The default of this parameter is 0%, referencing to unlimited soil fertility with the perfect concentrations of plant nutrients. Since this situation is very rare in real fields, even for well-maintained land, the value was manually tuned to 30% after initial model evaluation of daily biomass production with the CGLS-DMP (see section 3.1) product for several pixels. With this reduction in soil fertility, a good to moderate crop production over the entire domain can be simulated in absence of water stress, which is a setting recommended by expert knowledge of the AquaCrop source code developers.

A single crop file was created to simulate crop production over Europe. Spatial and temporal gaps of information at the ~1-km resolution prevent the inclusion of a more detailed crop parameterization. Furthermore, this research is focused on capturing relative temporal variation in biomass (not yield) for future use in a data assimilation system, a generic crop was developed and used for the entire domain. It is expected that regional differences of crop productivity from different crops will be corrected for via future data assimilation. After visual model evaluation and quantitative comparisons against satellite-based dry matter productivity (DMP, see below; Smets et al., 2019), the date of senescence was tuned manually, to optimally capture the length of the growing season. A generic reference crop was developed to simulate annual biomass development of C3 crops. C3 crops are predominantly found in temperate climates, as opposed to C4 crops that are more common in hot and dry climatological zones (Monfreda et al., 2008, Still et al., 2003). The crop was simulated as a transplant, assuming a small presence of vegetation from the start of the season, and with an annual cycle of 365 days, starting on the first of January. Because of this fixed annual cycle, the canopy development had to be simulated in calendar days instead of the more commonly used growing degree days. This results in an error in the simulation of canopy expansion during cold periods, but due to the consideration of growing degrees in the simulation of crop transpiration with the cold stress factor ($Ks_{Tr}$; Eq. 2), the reduced biomass production in these periods is still captured. As can been seen from equations 1 and 2, the factors that affect the crop development, simulated by canopy cover, are soil water stress and cold temperature stress. This generic crop file is mostly suitable to simulate canopy development during the spring and summer season. The main crop parameters are presented in table 1 and a flowchart of the model setup with input datasets is shown in Fig. 1.

**Table 1** Main crop parameters of generic crop to simulate biomass over Europe

| Generic crop main parameters | Input |
|---|---|
| Crop type | leafy vegetable crop |
| Crop is sown/crop is transplanted | crop is transplanted |
| Determination of crop cycle | calendar days |
| Coefficient for maximum crop transpiration ($K_{c,tr,x}$; -) | 1.10 |
| Base temperature (°C) below which crop development does not progress | 8.0 |
| Upper temperature (°C) above which crop development no longer increases with an increase in temperature | 30.0 |
| Minimum effective rooting depth (m) | 0.3 |
| Maximum effective rooting depth (m) | 1.0 |
| Normalized Water Productivity factor ($WP*$; g m$^{-2}$) | 17.0 |
| Calendar days from transplanting to recovered transplant | 0 |
| Calendar days from transplanting to maximum rooting depth | 80 |
| Calendar days from transplanting to start senescence | 232 |
| Calendar days from transplanting to maturity | 365 |
| Calendar days from transplanting to flowering | 0 |
| Minimum growing degrees required for full crop transpiration (°C - day) | 10.0 |

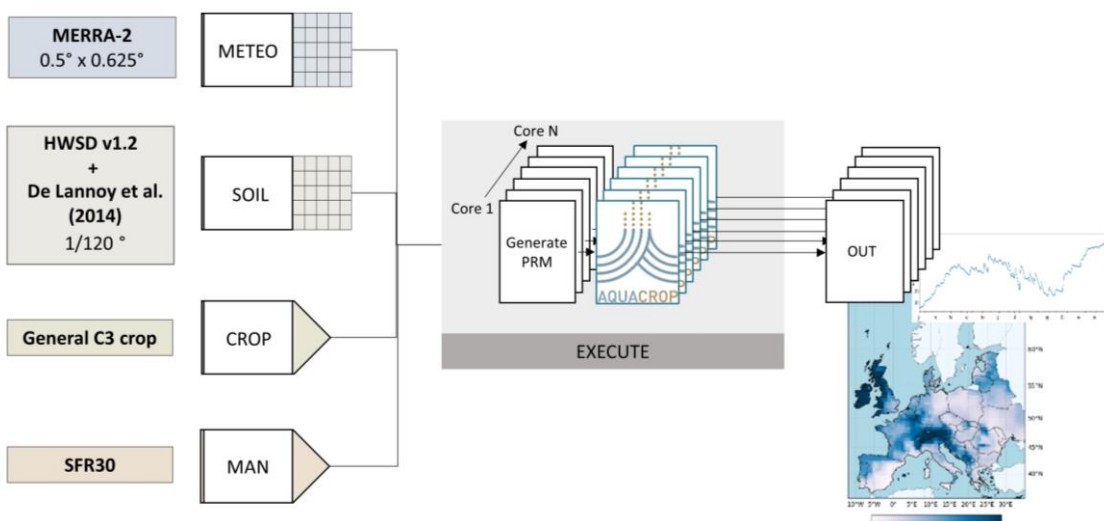

**Figure 1** Flowchart of the regional model setup with gridded meteorological and soil input data and generic crop and management input data indicated on the left side. The parallel computing system with a maximum of N cores can execute N pixels at the same time. The composited output files can then be visualized as maps or timeseries

## 3 Evaluation datasets and metrics

### 3.1 CGLS – DMP

To evaluate simulations of daily biomass production, the ~1-km dry matter productivity product from the Copernicus Global Land Service (CGLS-DMP; kg ha$^{-1}$ day$^{-1}$) was used (Smets et al., 2019). The CGLS-DMP is based on a simplified Monteith (1972) approach that makes use of the fraction of absorbed photosynthetically active radiation (fAPAR), which is derived from the optical satellite missions Satellite Pour l'Observation de la Terre (SPOT; 1999-2014) and Project for On-Board Autonomy - Vegetation (PROBA-V; 2014-June 2020), ECMWF re-analysis estimates of atmospheric variables such as radiation and temperature, and land cover information from the ESA CCA Land Cover Map. The retrieval algorithm is thus driven by atmospheric water availabilities, without explicitly accounting for water storage in the soil. The CGLS-DMP product is provided in 10-daily time steps, where each value is representative of the past 10 days for the years 1999 up to present date. To compare the data with the AquaCrop biomass, the nearest-neighbour function was used to spatially match the gridded simulations to the grid of CGLS-DMP and the median of the modelled daily biomass production was computed for the corresponding 10-daily intervals of the CGLS-DMP products. Since the crop parameterization in AquaCrop is suited to simulate the main growing season, the months November up to February were not included for the biomass evaluation.

### 3.2 CGLS – SSM

AquaCrop surface moisture content, i.e. the output of soil moisture in the top compartment of the soil profile, was evaluated with the CGLS relative surface soil moisture product CGLS-SSM. CGLS-SSM provides data for the top few centimetres of the soil, available at the same ~1-km resolution as CGLS-DMP. This product is derived from the C-band radar onboard Sentinel-1, processed by the TU Wien (Bauer-Marschallinger et al., 2018), and available from October 2014 onwards. Processing steps included geo-correction, radiometric calibration and normalization of the incidence angle. No correction was included for dynamics in vegetation or surface roughness. The data are provided as relative soil moisture estimates (%), that have to be multiplied by the porosity ($\theta_{sat}$) to convert to absolute volumetric soil moisture contents (m$^3$ m$^{-3}$). The Sentinel-1 satellite has varying overpass densities, resulting in a slightly different number of data-points in various areas, but the temporal resolution is generally between 3 to 8 days. To exclude potential datapoints for days in which the soil was nearly frozen, the soil temperature variable from MERRA-2 was used to identify and remove all data at which the soil temperature was below 4°C, following the recommended data masking by e.g. Gruber et al. (2020). The CGLS-SSM product contains masks for areas where it cannot be applied, i.e. a water mask, for pixels containing only water, a sensitivity mask, for pixels with a low sensitivity (urban, rivers, dense forests) and a slope mask, screening out pixels with a topographic slope higher than 17°.

### 3.3 SMAP – SSM

Surface soil moisture simulations were further evaluated with retrievals from the NASA Soil Moisture Active Passive (SMAP) mission, from April 2015 onwards. More specifically, the enhanced level-2 radiometer half-orbit, version 4, was used at 9-km

resolution (O'Neill et al., 2020, Chaubell et al., 2020). The SMAP radiometer measures L-band brightness temperatures in vertical and horizontal polarization at an incidence angle of 40°. It scans the earth's surface in a sun-synchronous orbit, which is 6:00 A.M. for descending and 6.00 P.M. for ascending mode, and with a temporal resolution of 2-3 days. The SMAP product provides three estimates of surface (~5 cm) soil moisture ($m^3\,m^{-3}$), derived from different retrieval algorithms (O'Neill et al., 2020). The 'Single Channel Algorithm using vertical polarization' is the current baseline for SMAP soil moisture and was also

used for AquaCrop evaluations.

SMAP data are projected onto the 9-km EASE grid version 2.0 (EASE2, Brodzik et al., 2012) and the AquaCrop soil moisture output was aggregated to this grid, by simply averaging all ~1-km pixels belonging to the same EASE2 grid cell. Only cells that were at least 50% filled with AquaCrop output were included for evaluation. The number of AquaCrop pixels per 9-km grid cell varies, depending on the location on the EASE2 grid. For SMAP-SSM, the recommended conservative quality control

was applied, and a temperature threshold of 4°C, derived from the GMAO GEOS land surface model, was applied to exclude nearly frozen soils (O'Neill et al., 2018).

### 3.4 In situ – SSM

In situ soil moisture measurements up to 5 cm depth were taken from the International Soil Moisture Network (ISMN, Dorigo et al. 2011) to evaluate AquaCrop simulations and satellite soil moisture products across all of mainland Europe. The

corresponding soil temperature data were used to exclude the dates with temperatures below 4°C. Whenever multiple in situ observation points were available within one AquaCrop pixel, the mean of those points was taken. AquaCrop simulations were crossmasked with both in situ data and the respective satellite product (CGLS-SSM, SMAP-SSM) to perform an in situ (and satellite-based) evaluation at each point. In situ data from the Hydrological Open Air Laboratory (HOAL) in Petzenkirchen, Austria (~ 49°57'N, 14°52'E) were made available by partners of the SHui consortium and contributed data to three extra

clustered pixels for CGLS-SSM, resulting in a total of 45 evaluation points for CGLS-SSM and 32 for SMAP-SSM in non-irrigated agricultural areas.

### 3.5 Metrics

To assess the temporal performance of the AquaCrop model, the bias, root mean square difference (RMSD), unbiased RMSD (ubRMSD), temporal Pearson correlation (R), anomaly correlation (anomR), were calculated with satellite and in situ products,

as follows:

$$Bias = \frac{1}{N}\sum_{n=1}^{N}(x_n - y_n) \tag{5}$$

$$RMSD = \sqrt{\frac{1}{N}\sum_{n=1}^{N}(x_n - y_n)^2} \tag{6}$$

$$ubRMSD = \sqrt{RMSD^2 - bias^2} \tag{7}$$

$$R = \frac{\frac{1}{N}\sum_{n=1}^{N}(x_n - \bar{x})(y_n - \bar{y})}{\sqrt{\left(\sum_{n=1}^{N}(x_n - \bar{x})^2\right)\left(\sum_{n=1}^{N}(y_n - \bar{y})^2\right)}} \tag{8}$$

where x are the simulated output data from AquaCrop and y are the observations from the satellite products and N are the number of observations. $\bar{x}$ and $\bar{y}$ are the time mean values. For the anomR, x and y are anomaly time series.

Comparing products with different spatial resolutions will always result in representativeness bias, which is especially acute when using in situ observations to evaluate pixel-scale estimates. Therefore, the focus of the evaluation was on temporal variability, using the R, anomR and ubRMSD metrics. The time period used for validation depended on the evaluation product.

When using satellite-based soil moisture, only grid cells were included when at least 150 CGLS-SSM or 200 SMAP-SSM retrievals (after quality control) were available during the overlapping period of satellite data (starting in 2014 for CGLS-SSM and in 2015 for SMAP-SSM) and simulations. When further comparing the satellite products to in situ data, a relaxed minimal threshold of 100 data pairs was set for the period of available data for each satellite product. For CGLS-DMP, the 10-daily data are complete between 2011-2018 and only March through November are included in the evaluation metrics.

The anomR was computed to assess both the short-term and inter-annual variability of biomass and soil moisture compared to the satellite products only, for lack of sufficiently long records of in situ data. A multi-year climatology (8 for CGLS-DMP, 4 for CGLS-SSM and 3.5 for SMAP-SSM) was computed and subtracted from the datasets to obtain anomalies as described by Gruber et al. (2020). The climatology is built on 31-day moving window averages, requiring either a minimum of 3 10-daily CGLS-DMP estimates or a minimum of 10 instantaneous CGLS-SSM and SMAP-SSM observations within a 31-day window.

The climatology of AquaCrop was computed using the same moving window and time period as the evaluation product. For surface soil moisture, only daily model output that matched the days of observations of the evaluation product was used, whereas for biomass evaluations, the data consisted of the median of the matching 10-day period.

In this study, only rainfed agriculture is considered. However, it is very likely that irrigation will occasionally take place on rainfed fields, where the timing and volume is based on local decisions made by farmers. Irrigation practices were not included
in the model simulations. To analyse how this human-driven process could influence the model performance, the FAO map 'Area Equipped for Irrigation' (AEI: Siebert et al., 2015), was used to identify areas that are occasionally irrigated and which were not necessarily captured by the irrigation class from the CORINE land cover inventory. The latter only considers regularly irrigated areas to distinguish rainfed from irrigated land. The available 1-km and 10-km AEI map version were used to stratify correlation values with CGLS-DMP and with SMAP-SSM, respectively.

**4 Results and discussion**

**4.1 Biomass**

A visual comparison of simulated and satellite-based biomass at different days in the year of 2017 is presented in Fig. 2 and gives an indication of the spatial performance of the regional AquaCrop model against the CGLS-DMP product. The figure

shows that the model is able to capture large regional and temporal differences in biomass production, but the absolute values

can differ between CGLS-DMP and the model. The coarser resolution MERRA-2 climate input is visible in the blocky pattern of the AquaCrop biomass maps. For the days in June and July, simulations over most of Italy ceased to produce biomass, whereas the CGLS-DMP product shows spatial variability in productivity. Water stress in the simulations is putting crop production to a halt, which is not in agreement to the CGLS-DMP. This can be either caused by an overestimation of water stress by the model, unmodelled irrigation, or because the CGLS-DMP product does not account well for drought stresses.

Drought stress is indirectly included in the CGLS-DMP via the observed fAPAR, but could still lead to overestimations of DMP in drier periods (Smets et al., 2019).

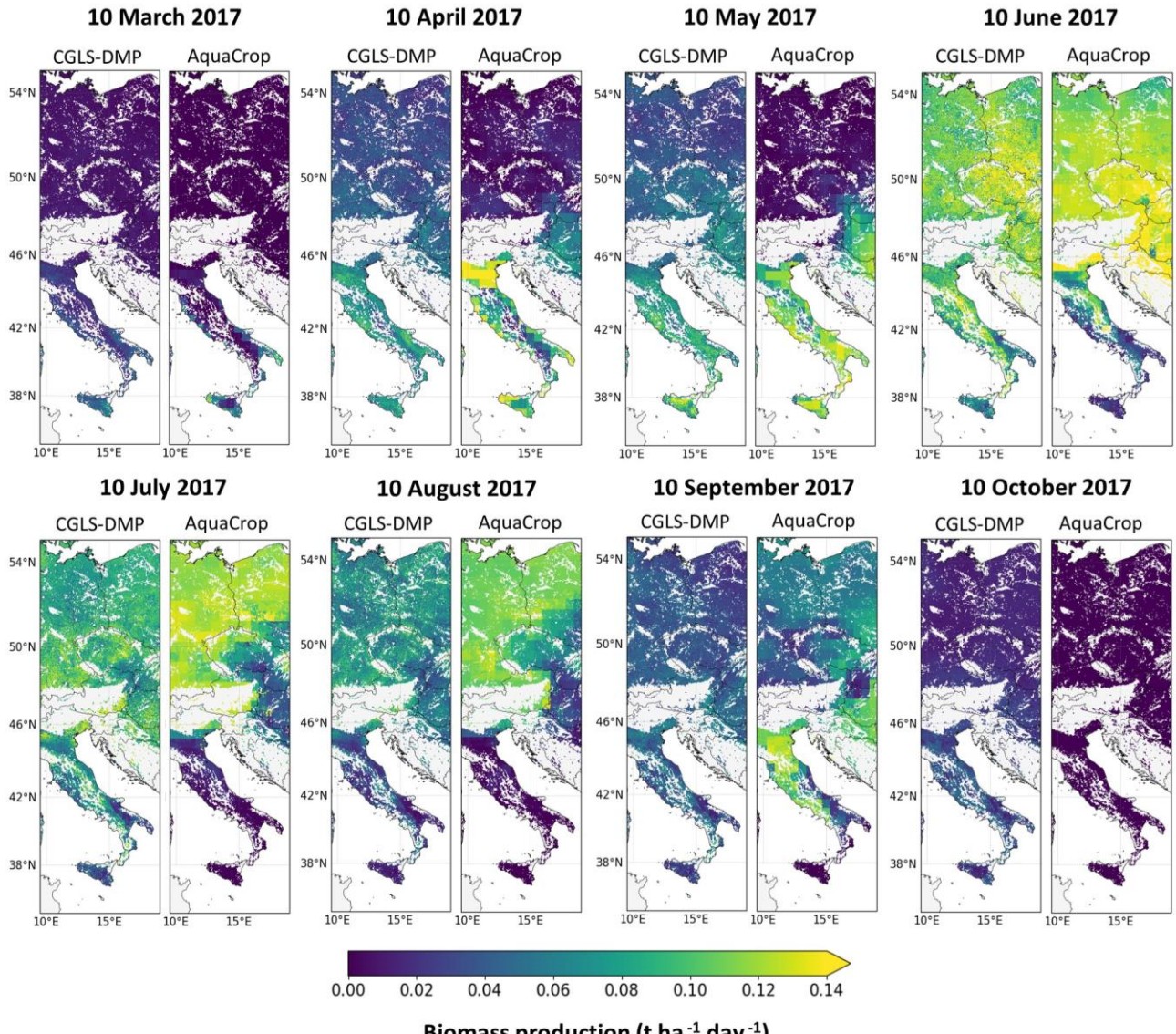

**Figure 2** Biomass production of CGLS-DMP and AquaCrop during different days of the year 2017. Light grey areas represent no data.

Figure 3 summarizes the performance metrics of AquaCrop biomass simulations against CGLS-DMP. Differences in absolute values of biomass estimates are inevitable, because of representativeness errors in both the model and satellite retrievals. For example, the model uses a generic crop, for which the parameters could be locally optimized. Nevertheless, the long-term biases are limited and cancel out over the entire domain. When focusing on the temporal variability, the temporal correlations indicate a high performance, with an overall mean of R=0.8. Higher correlations are mostly found in the northern part of Europe. Lower correlations are specifically found in the upper North and in the South (Italy). Similarly, the ubRMSD is highest in the southern half of the study domain. The spatial variability in ubRMSD can be attributed to different factors that limit crop growth, which will be mostly cold temperatures in the North, and low soil water contents in the South. Across the domain the ubRMSD is 0.03 t ha$^{-1}$ day$^{-1}$ and typically less than 20% of the amplitude in biomass production. The anomaly correlation is lower than the correlation, but still significant, with a mean of anomR=0.46. The raw correlation includes the trivial agreement in the seasonal variability and is thus inevitably higher, whereas the anomaly correlation only evaluates short-term and interannual variability, as illustrated in Fig. 5 for the HOAL catchment in Austria. Both the model and satellite data show anomalous high biomass production in June 2017, whereas anomalous low values are found in both datasets in June 2013. The short-term anomaly biomass productivity increments are also corresponding well to the evaluation data, but for AquaCrop they are often more pronounced. For regions in the South of Europe (Italy), simulated productivity anomalies are much more pronounced, clearly showing the modelled response on stronger rainfall events after a relatively dry period. When comparing this to the CGLS-DMP product, it shows anomalies that are either less extreme or do not match the anomalies of the model simulations, resulting in lower anomaly correlations (Fig. 5). This emphasizes the importance of high-resolution precipitation information for climatic regions in which precipitation is the main limiting factor for crop production. Across the northern region, the lower anomaly correlation values can be partly associated with soil texture (TAW) as can be seen from Fig. 3 and 4. In areas where there is a sufficient amount of rainfall, but soils are typically sandy and have a low TAW and high $K_{sat}$, soil water easily drains through the profile, which prevents optimal crop production. The effect of such stresses may not be observed in the CGLS-DMP, and will result in deviating interannual variabilities.

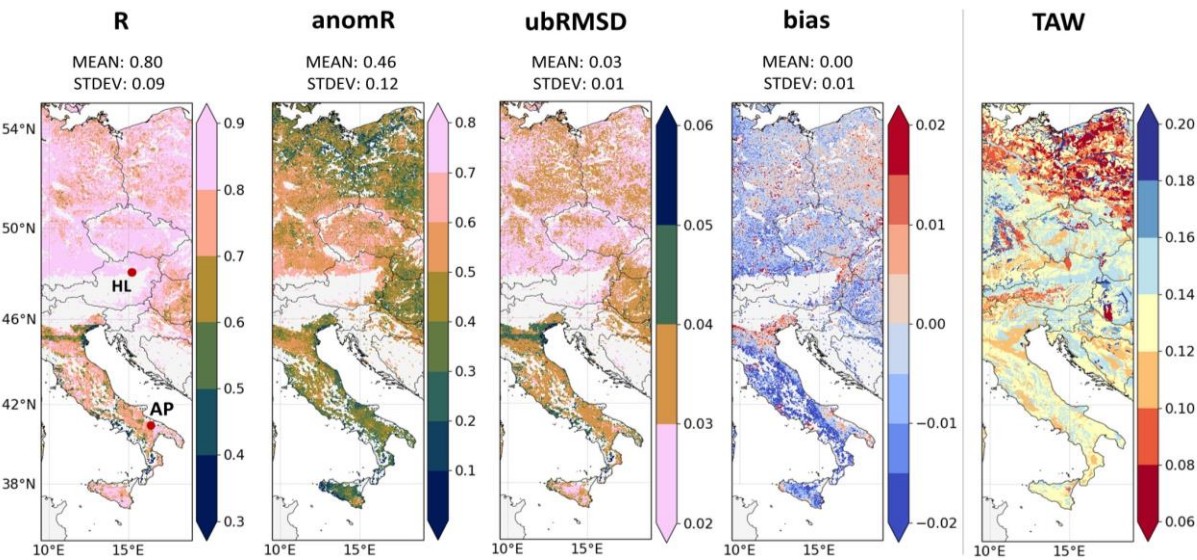

**Figure 4** Temporal metrics of AquaCrop biomass evaluated against CGLS-DMP, with metrics R (-), anomR (-), bias (t ha day$^{-1}$), ubRMSD (t ha day-1). Spatial mean and standard deviation of the metrics are indicated with MEAN and STDEV. Also shown is the TAW (m$^3$m$^{-3}$) computed as the field capacity minus wilting point, without taking rooting depth into account. Light grey areas represent no data. Boxplots showing the distribution of biomass anomaly correlations for the northern half of the study domain (46°N – 55°N), grouped by different TAW ranges

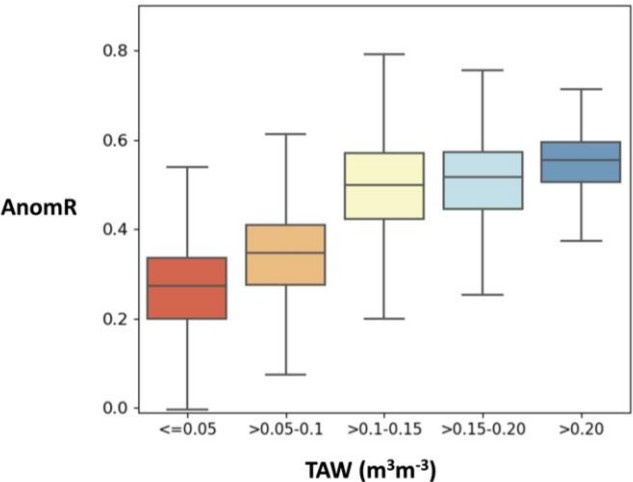

**Figure 3** Boxplots showing the distribution of biomass anomaly correlations for the northern half of the study domain (46°N – 55°N), grouped by different TAW ranges.

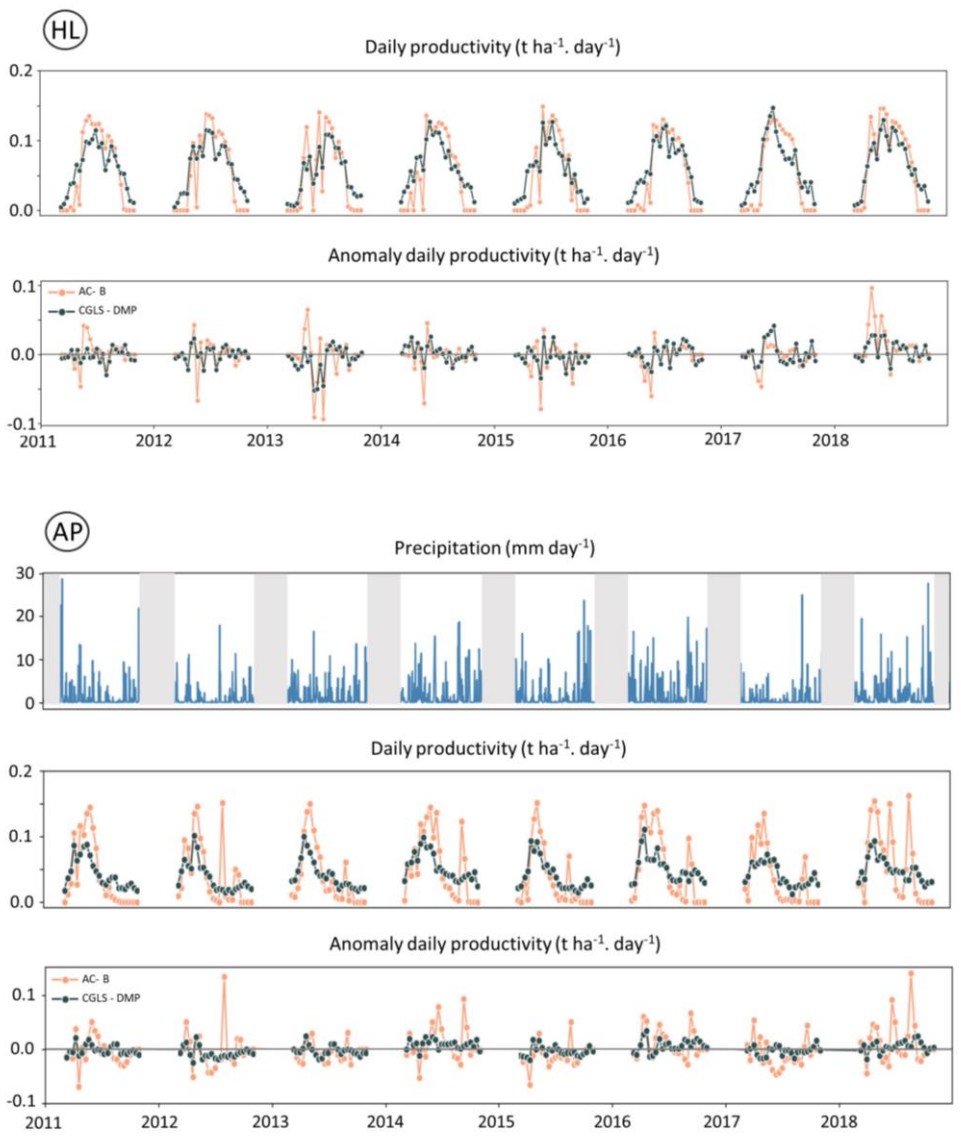

**Figure 5** Time series of biomass productivity, anomaly daily productivity (and precipitation) for (HL) the HOAL catchment in Austria, and (AP) a pixel in the Apulia region of South Italy, both marked in Fig. 3. Precipitation is only shown for AP, because it has a marked effect there on short-term anomaly productivity. Periods between October and March are masked out in grey for precipitation

## 4.2 Surface moisture content

Surface soil moisture content was evaluated using three products at different scales; point measurements from ISMN and some
additional sites in the HOAL catchment, 1-km CGLS-SSM and 9-km SMAP-SSM. Figure 6 shows the AquaCrop performance
metrics against the satellite data. The spatial mean R and anomR value with SMAP retrievals are 0.74 and 0.65, respectively.
The anomR is especially high in the central part of Europe and decreases towards the North. Overall, AquaCrop is much better
correlated with SMAP-SSM than with CGLS-SSM. The mean R and anomR value of AquaCrop SSM with CGLS-SSM are
0.52 and 0.50, respectively. Figure 7 illustrates that the lower agreement between AquaCrop and CGLS-SSM data is not solely
due to the inevitably higher noise in the finer-scale CGLS-SSM data. When aggregating CGLS-SSM to the EASE2 9-km grid
using the same spatial mask of SMAP-SSM, the temporal correlations with AquaCrop increase slightly, with a mean R of 0.57
(spatial standard deviation STDEV: 0.08) and a mean anomR of 0.56 (STDEV: 0.06) (Fig. 7), and remain well below the
correlation values between SMAP-SSM and AquaCrop SSM.

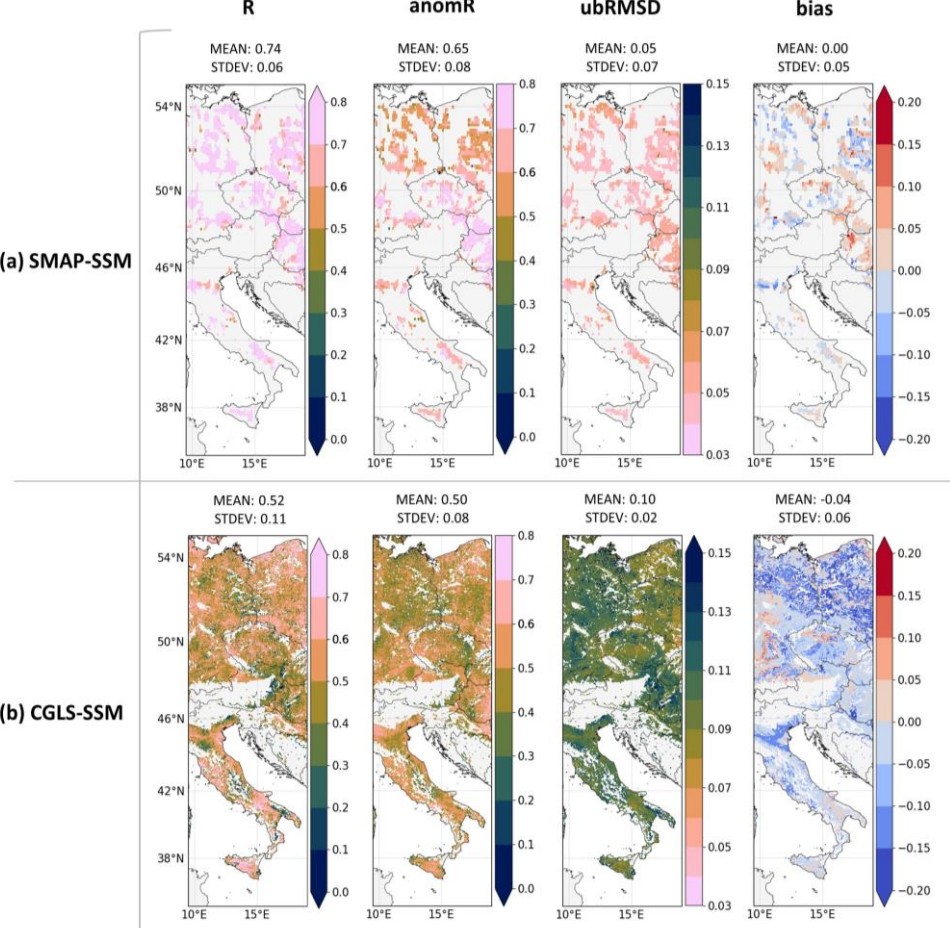

**Figure 6** Temporal performance metrics of AquaCrop SSM evaluated against (a) SMAP-SSM and (b) CGLS-SSM, i.e. R (-), anomR (-),
bias ($m^3\ m^{-3}$) and ubRMSD ($m^3\ m^{-3}$), with indication of the spatial mean and standard deviation of the metrics (MEAN, STDEV). Light grey
areas represent no data.

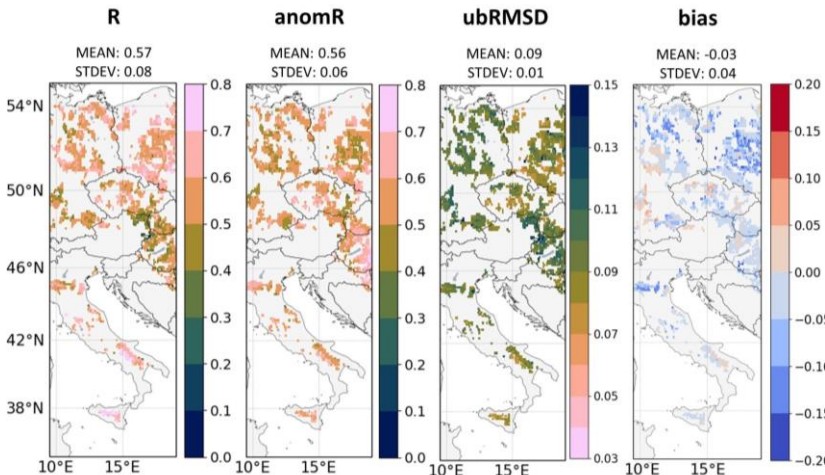

**Figure 7** Same as Figure 6b, but after aggregation of the CGLS-SSM data to the 9-km EASEv2 grid and spatial crossmasking with SMAP-SSM data.

Several areas with higher elevations have lower correlation values (central Italy, eastern Alps). The spatial correlations of
AquaCrop SSM on the 9-km EASE2 grid with 9-km CGLS-SSM and 9-km SMAP-SSM reveal a large variability in time, with a temporal mean spatial R of 0.38 and temporal standard deviation of 0.21 for CGLS-SSM and a mean R of 0.32 and temporal standard deviation of 0.22 with SMAP-SSM.

When looking at the absolute values of the bias and ubRMSD, the evaluation of AquaCrop against CGLS-SSM (1-km or 9-km) is also far worse than that against SMAP-SSM, but the spatial pattern of the errors is similar for SMAP-SSM and CGLS-
SSM. The spatial mean ubRMSD against SMAP-SSM is 0.05 $m^3 m^{-3}$, close to the global target product uncertainty of 0.04 $m^3$ $m^{-3}$ (Entekhabi et al., 2014), and the spatial mean ubRMSD against 1-km CGLS-SSM is 0.10 $m^3 m^{-3}$. Also here, the effect of soil texture on model performance was found. The ubRMSD values of 0.14 $m^3 m^{-3}$ and higher for 1-km CGLS-SSM, correspond to outliers exactly to a specific soil class in the HWSDv1.2 classification, that contains 93% sand. This soil class is characterised by very high $K_{sat}$ and very low values for $\theta_{WP}$ and $\theta_{FC,}$ resulting in extremely low simulated available moisture
content in the top layers. Because the low $\theta_{WP}$ is very close to the soil evaporation demand, the model is not able to simulate soil moisture correctly for the top layers for daily timesteps. AquaCrop is a crop simulation model and this soil class is unrealistic for agricultural land. In future applications when multiple datasets from different sources are combined, it is recommended to limit the simulations to possibilities that are actually suitable for the specific simulation purpose. Nonetheless, the poorer performance against the 1-km Sentinel-1-based CGLS-SSM is in general not due to model shortcomings, but
dominated by poor satellite retrievals, as will be discussed below.

A comparison between in situ data, 1-km CGLS-SSM, 9-km SMAP-SSM and 1-km AquaCrop surface soil moisture at ISMN sites and 3 sites in the HOAL catchment is shown in Fig. 8. Across the in situ sites, the mean R value between AquaCrop and in situ soil moisture is 0.61 (Fig. 8a) and higher than the mean R value of 0.52 with CGLS-SSM (Fig. 8b). The mean ubRMSD between AquaCrop and in situ measurements is 0.06 $m^3 m^{-3}$, significantly lower than the mean between AquaCrop and CGLS-

SSM (0.10 m$^3$ m$^{-3}$). The mean R between Sentinel-1 CGLS-SSM and in situ data is even lower, with a value of 0.42 and a mean ubRMSD of 0.11 m$^3$ m$^{-3}$ (Fig. 8c). The comparison with the satellite products over in situ sites shows that SMAP-SSM mean temporal correlations are significantly better with both AquaCrop simulations (Fig. 8b; R=0.81, ubRMSD=0.05 m$^3$ m$^{-3}$) and in situ measurements (Fig 8c; R=0.69, ubRMSD= 0.05 m$^3$ m$^{-3}$) than CGLS-SSM, even though SMAP-SSM has a lower spatial resolution. This is further illustrated in the time series at three locations presented in Fig. 9, where SMAP-SSM follows

the pattern of in situ data well and slightly better than AquaCrop, whereas the pattern of the CGLS-SSM values is more erratic. The high correlations between SMAP-SSM and in situ measurements show that SMAP-SSM is better at capturing variations at smaller scales than the current system of AquaCrop, due the coarse resolution of meteorological input data. Additionally, SMAP-SSM retrievals probably benefit from a more accurate background representation of the vegetation, whereas AquaCrop uses a generic crop description. For CGLS-SSM, lower observed soil moisture was often found for the months April, May and

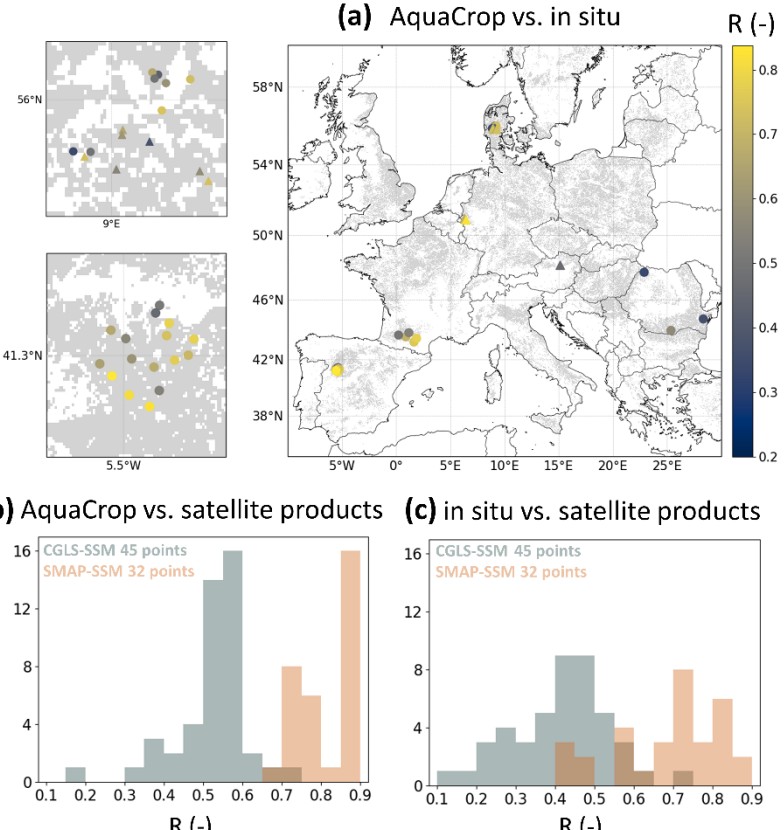

**Figure 8** (a). Pearson correlation R values between in situ measurements from the ISMN and AquaCrop surface soil moisture, at 45 locations over Europe, with grey pixels containing at least 50% rainfed agriculture according to the CORINE land cover map 2012. Correlations shown are from crossmasked data with CGLS-SSM. The circles indicate the locations used for both evaluation with CGLS-SSM and SMAP-SSM, whereas triangles show locations that were only used for CGLS-SSM. (b) Histogram of R values between AquaCrop surface soil moisture and the two satellite products CGLS-SSM (45 points) in grey and SMAP-SSM (32 points) in orange, at the locations of the in situ sites. (c) Histogram of the R values between the in situ measurements and the two satellite products CGLS-SSM (45 points) in grey and SMAP-SSM (32 points) in orange.

June, as can be seen in Fig. 9b and c. The poor correlation of CGLS-SSM during these months is most likely due to the fact that the Sentinel-1 backscatter signals are dynamically affected by changing vegetation during the growing season, but the soil moisture retrievals are only corrected for with a static vegetation value for every day of the year. Furthermore, changes in surface soil roughness are not accounted for in the retrievals and could play an important role in the lower quality of the CGLS-SSM retrievals (Bauer-Marschallinger et al., 2018).


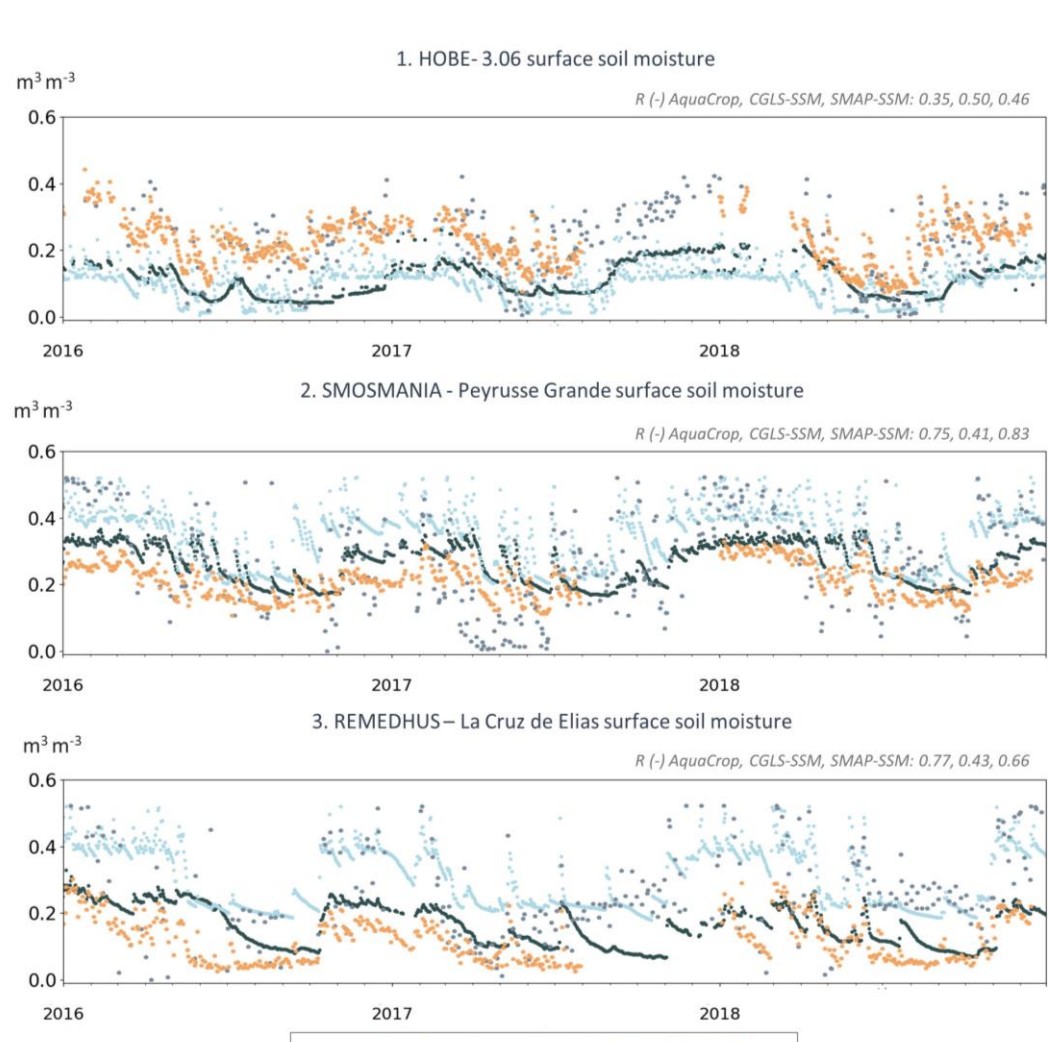

**Figure 9** Time series of daily surface soil moisture at three locations marked in Fig. 6a: 1 (~ 55°54 N,8°52 E), 2 (~ 41°17 N, 5°18 W) and 3 (~ 43°39 N, 0°13 E): AquaCrop (light blue) in situ measurements (dark grey), CGLS-SSM (light grey) and SMAP-SSM (orange). Pearson correlations R of in situ data with the different products are given for each location.

## 4.3 Effect of irrigation

Figure 10 shows the spatial distribution of the R values of AquaCrop biomass and soil moisture with CGLS-DMP and SMAP-SSM, respectively, grouped into two percentage classes of AEI. In terms of biomass, higher R values between AquaCrop and CGLS-DMP (mean R= 0.81) are found for pixels where AEI < 10% than for areas where AEI >= 10% (mean R= 0.72). For soil moisture, the correlation with SMAP-SSM shows barely any difference between the AEI groups (AEI < 10%: mean R = 0.74; AEI >= 10%: mean R= 0.73). It should be noted that SMAP-SSM has a much smaller coverage than the CGLS-DMP, because SMAP-SSM is screened conservatively based on its quality flags. The results of this comparison suggest that, even if the simulations were limited to dominantly rainfed agricultural areas according to the CORINE land use map and therefore did not include irrigation, it is possible that in reality irrigation is occasionally applied in rainfed fields and seen by the satellite data, resulting in lower correlation metrics.

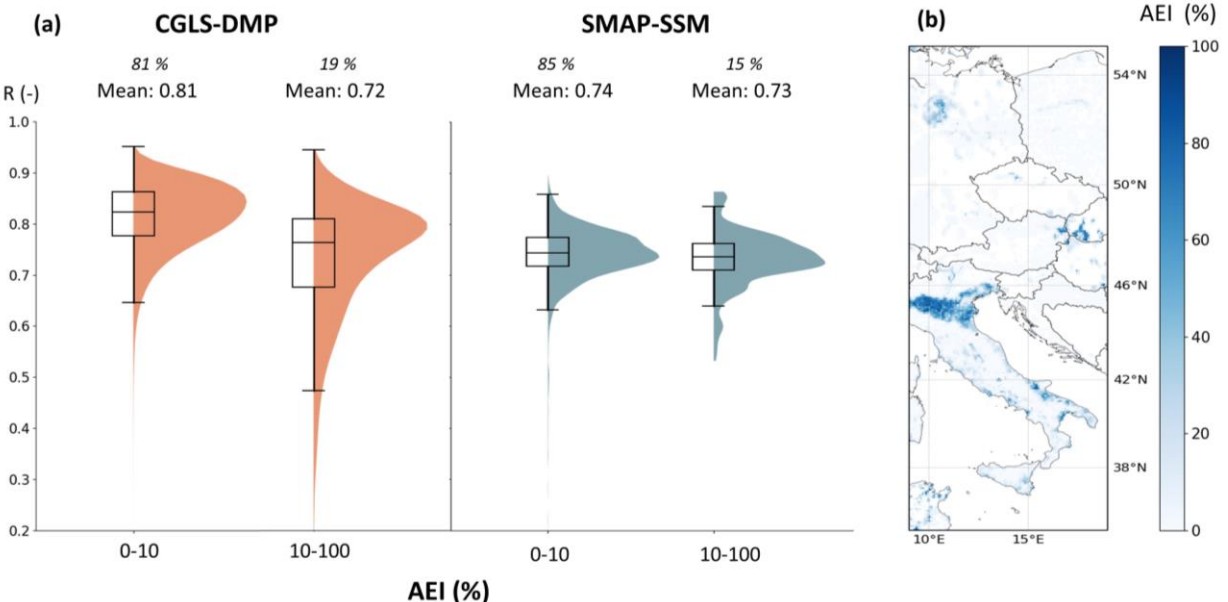

**Figure 10** (a) Boxplots with violin curve of temporal R values grouped by FAO's percentage of Area Equipped for Irrigation (AEI), group1: 0-10% and group2: 10-100%, on the left side for CGLS-DMP and on the right side for SMAP-SSM. The percentage of the total amount of pixels for each group, and the spatial mean R value is noted at the top of the figure. (b) AEI map over study domain.

## 4.4 Discussion of the regional AquaCrop model

The current gridded AquaCrop model has several conveniences, such as the efficient parallel processing structure, the ability to run at any resolution and domain, and the modular setup in which a compiled executable can be easily replaced by newer

AquaCrop versions. The model setup is chosen to facilitate subsequent embedding within a future satellite-based data assimilation system.

The regional modelling system was designed to capture the seasonal and inter-annual variability, with some important simplifications. A general C3 crop was assumed, and management data was considered as homogeneous over the entire study area, whereas meteorology and soil information were spatially variant. Therefore, the evaluation of this regional crop model setup against satellite products was mainly done in terms of unbiased temporal metrics. AquaCrop accurately simulates the temporal variability in biomass and surface soil moisture, especially in the northern regions and if the soil's TAW is not limiting. Limitations in the accuracy of the input precipitation (MERRA-2) causes slightly worse simulations in the water limited southern regions, where biomass shows a fast response to limited (but sometimes inaccurately timed) rainfall events. The use of high-resolution meteorological forcing is likely to be most important next step to further improve fine-scale AquaCrop simulations. The evaluation was limited to surface soil moisture and biomass, but could be further expanded to other variables such as root-zone soil moisture and transpiration in the future. Reference data for the latter variables are always informed by strong (often modelled) background information (Martens et al., 2017, Reichle et al., 2019) and not directly observed over large regions. Furthermore, applying crop specific parameters to the crop file would most likely result in better biomass and yield simulations, which would mainly improve the temporal bias and spatial performance metrics.

The suitability of this modelling system to estimate the spatial variability in soil moisture and yield production for specific crop types would require further analysis and more detailed input information. For example, by combining input datasets from different sources, some unsuitable cropland areas were identified (e.g. too low TAW in combination with high $K_{sat}$) that were not filtered out from this analysis. Furthermore, unmodeled irrigation could influence the regional model performance. Most importantly, the relative spatial variability in biomass is likely not dominated by meteorology and soil texture, but by the various types of crops. The parameters associated with each of these crops could be spatially optimized (calibration, data assimilation for parameter estimation) in future work, using historical time series of spatially covering reference data, e.g. optical Sentinel-2 data.

The regional model evaluation could only be performed with satellite retrievals, but such an evaluation is limited to the days of overpass, and to times and locations where retrievals are of sufficient quality. For example, SMAP-SSM retrievals are filtered out under too dense vegetation or frozen conditions. Furthermore, the satellite signal may represent a slightly different quantity than what is modelled. Additionally, microwave signals only pertain to the upper 5 cm of the soil, but the model's surface layer is 10 cm. The provided quality flags on CGLS-SSM are less strict, providing a better spatial coverage of fine-scale data. However, the C-band soil moisture measurements pertain to an even shallower soil depth and are likely more affected by vegetation. In any case, both the satellite retrievals and model simulations have their own systematic and random errors. The influence of the former is suppressed in this study by focusing on relative temporal variability. To further dynamically improve model simulations, or to add value to the available satellite data (e.g. dynamically interpolate) via AquaCrop modelling, random errors in both sources can be limited via data assimilation for state updating.

## 5 Conclusions

In this paper, a spatially distributed version of the field-scale AquaCrop model v6.1 is presented and evaluated against various
satellite data products and in situ data. The new regional AquaCrop infrastructure allows to simulate biomass and soil moisture
over large domains in an efficient way, due to the massive parallelization of the gridded simulations. In this case study, the
regional AquaCrop model is forced with meteorological input based on MERRA-2 re-analysis data, the soil information is
extracted from the HWSDv1.2, and a generic crop is parameterized. Even when using coarse meteorological input data, the
AquaCrop model can capture seasonal, interannual and short-term temporal dynamics of biomass over Europe at a fine ~1-km
resolution. For the years 2011 through 2018, the temporal R between the AquaCrop biomass production and CGLS-DMP is
0.8, and the anomR is 0.46, across central Europe. The R values are higher in the northern half of the study domain, where
crop growth is generally temperature limited, whereas in the southern half of the domain, water stress becomes more important
and the R values are lower. Likely factors that can influence this difference in correlation are an underrepresentation of drought
stress by the CGLS-DMP product, the effect of occasionally applied irrigation which is not included in the model, or possibly
overestimations of simulated drought stress by the model. Additionally, the impact of soil parameters is apparent in the anomR
values, where lower TAW values in the northern part result in differing anomalies for modelled biomass and CGLS-DMP.
The AquaCrop simulations for surface moisture content show that seasonal, interannual and short-term temporal dynamics
correspond well to the 9-km SMAP-SSM data, with a mean R value of 0.75 and an anomR value of 0.65 across the study
domain. Lower R values are found for Sentinel-1 CGLS-SSM, with a mean temporal R of 0.52 (aggregated to 9-km EASE2
grid: 0.57) and a similar anomR of 0.50 (aggregated to 9-km EASE2 grid: 0.56). The comparison between AquaCrop, CGLS-
SSM, SMAP-SSM and in situ data for 45 (32 for SMAP-SSM) locations in Europe shows that both AquaCrop and SMAP-
SSM better agree with in situ data (mean R= 0.61, 0.69, respectively) than Sentinel-1 CGLS-SSM (mean R= 0.52). The lower
performance of Sentinel-1 CGLS-SSM can be attributed to the static correction for vegetation, which causes soil moisture
retrieval errors during the growing season, and the fact that there is no correction for surface roughness (Bauer-Marschallinger
et al., 2018). For both the evaluations with SMAP and Sentinel-1 retrievals, the effect of soil characteristics influences the
evaluation performance of the AquaCrop model. When certain soil characteristics are unsuitable for crop cultivation, such as
a high $K_{sat}$ and a very low $\theta_{WP}$ and low TAW, soil moisture becomes inaccurately represented by the AquaCrop model,
increasing the model error. At the same time, satellite-based soil moisture retrievals also contain errors related to a priori
defined soil hydraulic parameters.

Improvements to the regional AquaCrop model can be made in terms of higher resolution meteorological input data to better
capture small-scale spatial differences, by revising the soil hydraulic parameters to better represent soil types used for
agricultural land, and by introducing spatio-temporally varying crop parameters when such information becomes available.
Overall, the current model is able to well represent temporal and spatial differences at the field and regional scale in both
biomass production and surface soil moisture, requiring only easily accessible input data. The computationally efficient
modelling system is ideal to foster future improvements in the spatial patterns in both soil moisture and biomass production

via local parameter optimization based on historical records of satellite data, and improvements in the short-term and interannual temporal variations via sequential satellite data assimilation.

*Code and data availability*. The code and data needed to run the regional version of AquaCrop v6.1 on a Linux-based system is available on Zenodo, with DOI: 10.5281/zenodo.4770738. Apart from the code, this repository includes the generic crop
file, the management file and ancillary soil data from De Lannoy et al. (2014). All other input data and evaluation datasets are freely available, except for the in situ measurements of the HOAL experiment site. Please visit the following links for data access: MERRA-2 variables (accessed on March 2019): https://disc.gsfc.nasa.gov/datasets?project=MERRA-2; the soil mineral classification and organic matter originates from HWSDv1.2: http://www.fao.org/soils-portal/data-hub/soil-maps-and-databases/harmonized-world-soil-database-v12/en/; the CORINE land cover map (accessed on September 2019):
https://land.copernicus.eu/pan-european/corine-land-cover; evaluation datasets from COPERNICUS Global Land Service (CGLS-DMP, CGLS-SSM; accessed on February 2020, June 2020): https://land.copernicus.eu/global/themes/vegetation; SMAP Enhanced L2 Radiometer Half-Orbit 9 km EASE-Grid Soil Moisture, Version 4 (accessed on September 2020): https://nsidc.org/data/SPL2SMP_E/versions/4; ISMN soil moisture at 5 cm depth (accessed on August 2020): https://ismn.geo.tuwien.ac.at/en/; FAO irrigation maps (accessed on June 2020): http://www.fao.org/aquastat/en/geospatial-
information/global-maps-irrigated-areas.

*Author contributions*. SDR created the code to execute the regional version of the model, prepared the input data and conducted the model evaluation. GDL prioritised the main steps taken in the paper, provided supervision and scientific guidance throughout all research advances and manages HPC usage. DR provided scientific guidance regarding the use and interpretation of the AquaCrop model, developed the generic crop file, and provided the source code of AquaCropv6.1. SDR
wrote the paper and all authors contributed.

*Competing interests*. The authors declare that they have no conflict in interest.

*Acknowledgements*. The authors would like to thank the HPC VSC team, in particular Geert-Jan Bex and Martijn Oldenhof, for their help during the AquaCrop compilation on the VSC HPC. We would also like to thank Peter Strauss and Gerhard Rab from Vienna University of Technology (TU Wien) for sharing their data from the HOAL experiment site, and Stefan Siebert
for providing a 1-km map with area equipped for irrigation. Finally, we greatly appreciate the review comments from Christoph Müller, an anonymous reviewer and the editors.

*Financial support.* This research is conducted as part of the H2020 project SHui, that stands for *"Soil Hydrology research platform underpinning innovation to manage water scarcity in European and Chinese cropping systems"*. SHui is co-funded by the European Union Project GA 773903 and the Chinese MOST.

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
