# Peer review of "Performance analysis of regional AquaCrop (v6.1) biomass and surface soil moisture simulations using satellite and in situ observations"

_Geoscientific Model Development, 2021_

## Author Response (AR1)

**Reviewer #1 (Christoph Muller):**

The authors would like to thank the reviewer, Christoph Müller, for his elaborative comments to improve our manuscript. We have considered each comment in the revised manuscript and would like to provide an overview of the adjustments.

**Upon request from the executive editors they have made available the source code of the wrapper and the input data, however the source code of the AquaCrop model is not published but only an executable file, not as the source code. This seems to violate the open access policies of GMD.**

Answer: The original source code is exclusively licensed by the Food and Agricultural Organization (FAO). The source code of the executable on our Zenodo link is equal to the source code of the AquaCrop windows programme version 6.1 (http://www.fao.org/aquacrop/software/aquacropstandardwindowsprogramme/en/), but compiled for a Linux operating system. We would like to emphasize that only the executable is needed to run this spatial version of AquaCrop.

**The manuscript lacks clarity in many cases (see detailed comments below) but also on the objective(s) of the paper. From the source code provided and the setup, it is meant to describe the parallel model framework and to evaluate model performance (not the parallel framework). However, the model performance is evaluated in the manner of individual points (despite that it's a quite large set of points), not in a manner that addresses the scale and extent, e.g. by addressing the ability to reproduce spatial patterns, which would be a main asset of "regional" model applications compared to a set of field-scale applications. If the objective is to present the framework that allows for parallel, high-resolution, large-scale applications, the technical skill and spatial properties of the simulation could have stronger emphasis in the evaluation, or how the large-scale setup (e.g. lack of calibration) compares to field-scale setup.**

**If (one of) the objective(s) is to generally evaluate AquaCrop against novel data (such as the data sets used here), the model description needs to be expanded, the current set of equations does not even address all processes discussed as relevant in the text (see comments below).**

Answer: The model is not only evaluated on individual points, but also regionally, using satellite data. As has been corrected now in the introduction, the emphasis of this study is on capturing temporal patterns and less attention has been paid to analyse spatial model performance. To further investigate the spatial patterns of the simulations, we now also added information about the spatial correlation values between regional AquaCrop soil moisture simulations and satellite retrievals. Note that this analysis is risky by itself because the absolute values of satellite retrievals depend themselves on local parameter estimates in the retrieval algorithm (that might not be anywhere close to the 'truth'). Similarly, the model setup with a generic crop is not meant to correctly estimate the 'absolute' values of biomass. Therefore the relative time series analysis at all pixels is deemed more important: our crop modelling system is slated by the state of the art practices in land surface modelling and data assimilation, where relative variability is much more important (e.g. anomalies) than absolute values.

Specific details to clarify the focus on temporal variability are provided in the detailed comments below, and the following text concerning spatial correlations has been added to the manuscript:

L 343-346:

*"Spatial correlations of AquaCrop SSM on the 9-km EASE2 grid reveal a large variability in time, with a temporal mean spatial R of 0.38 and temporal standard deviation of 0.21 for CGLS-SSM and a mean R of 0.32 and temporal standard deviation of 0.22 with SMAP-SSM."*

**I don't understand the claim made that AquaCrop could serve as a bridge between point and global level simulations. It is claimed that AquaCrop was developed for a simplistic representation of crop growth (L46) and performance is good if the model is calibrated for local field conditions (L48). So AquaCrop may actually lack processes that are relevant to capture the heterogeneity of the landscape of environmental and management conditions at larger scales. The calibration of large-scale applications is hampered by lack of data that could serve as calibration targets and is not attempted here.**

Answer: We agree that this statement cannot be confirmed with our study and have edited the text as follows:

L48-L49:

*"The flexible model setup will allow for many different applications, but in this study the focus is on the preparation of a satellite-based data assimilation system."*

**Even though an eyeball comparison suggests that results hold true, I find the comparison of the AquaCrop performance against the 2 soil wetness datasets a bit biased, as the samples are very different. This could be made more direct if also the statistics would be supplied for the set of pixels that is covered by both reference data sets.**

Answer: All our analyses and performance evaluations are based on a range of objective skill metrics, community-based standards and direct causal/physical relationships – no eyeball comparisons. We further specify this in our responses to the detailed comments below.

We are not entirely sure about the 'bias' in the performance analysis. However, based on suggestions below, we think that (i) there might have been some misunderstanding, which we will correct for in the text, and (ii) that a common spatial mask (crossmasking of datasets) is recommended. In our revised version, the CGLS-SSM is also aggregated to the SMAP EASE2 grid and an extra performance analysis is done for both satellite products with AquaCrop using the same spatial mask. To be able to keep a sufficient amount of data for both datasets, this crossmasking is only done over space, not in time. A crossmasking of datasets done in space and time would reduce our datasets to a very small overlapping sample, because each satellite dataset has very different recommended retrieval quality flags and overpass times. This would then result in a great loss of information and a consequent bias in our performance analysis (limited to a small subsample). Please see details below.
* * *
Detailed comments:

**L19: curious to learn about how that bridge could look like. Many globally applied crop models are, in fact, field-scale models run in a modeling framework to process gridded data**

Answer: Please see description above. We have removed this statement from the introduction.

**L31: the better GGCMI evaluation reference would be Müller et al. 2017, not 2018. You could add Folberth et al. 2019 (http://dx.doi.org/10.1371/journal.pone.0221862)**

Answer: Thank you for the references. They are both included in the revised manuscript.

**L34: this "downside" is not only relevant for upscaling field-scale models but holds true for any large-scale crop model application.**

**L35: do you mean "… and loss of information that is typically available at smaller scales"?**

Answer: Thank you, the text for L34 & L35 has been updated as follows:

L35-L36:

*"A downside of large-scale crop models, especially at a global level, is that they often suffer from the generalization of input data and loss of information that is typically available at smaller scales, resulting in larger errors at the local scale."*

**L39: I guess the better reference for the GGCMI Project per se is Elliott et al. 2015 (http://dx.doi.org/10.5194/gmd-8-261-2015) if you just want to have one reference. If you want to describe the breadth of crop models used in GGCMI, you could add Elliott et al. (2015), Müller et al. 2019 (http://dx.doi.org/10.1038/s41597-019-0023-8), Franke et al. (2020a) (http://dx.doi.org/10.5194/gmd-13-2315-2020) for the different set of models contributing to Phase2 and (Jägermeyr et al. under review) for yet another ensemble of crop models contributing to the current Phase 3 (including AquaCrop).**

Answer: Thank you for the references, Elliot et al. (2015) has been included in the manuscript.

**L40: odd second half of that sentence: response to what? Do you mean "… more insight is needed in the relevancy of different processes represented in crop models applied at different spatial and temporal scales under different management assumptions"?**

Answer: Thank you for the suggestion, the sentence has now been removed from the manuscript. A new section has been added to better clarify our motivation.

L39-L46:

*"A possibility to correct for scaling errors, is the updating of the model simulations with remote sensing observations via data assimilation. There are several studies that have already used data assimilation in regional crop modelling systems (De wit & van Diepen, 2007; Mladenova et al., 2019; Zhuo et al., 2019), either for parameter or state updating. Parameter updating or calibration allows to match the absolute values of the simulations with (most often historical) observations. State updating allows to correct the relative temporal evolution and to obtain better initial conditions for subsequent model predictions. To get the most optimal results with data assimilation, it is important to start with a reliable model that is able capture the seasonal as well as interannual temporal variabilities."*

**L52: unclear – what are the difficulties to update to newer AquaCrop versions?**

Answer: This statement has been removed from the sentence.

**L58: if this is the main objective, what are the side objectives?**

Answer: This was an unfortunate formulation. The evaluation of high-resolution regional AquaCrop simulations is *the* objective of this paper. This has been changed in the revised manuscript (L75).

**L60: I'm surprised by the "generic crop" approach. Crops differ in various aspects and e.g. differences in growing season specifications have been shown to matter quite a lot (Müller et al. 2017, Jägermeyr & Frieler 2018 (http://dx.doi.org/10.1126/sciadv.aat4517)). Also above ground biomass differs substantially between crops. I see that this liberates you of having to specify high-resolution**

**inputs that match crop distributions that the satellites see, but some justification of this choice would be needed here.**

Answer: As was briefly mentioned at the end of the first manuscript, this model is set up for satellite-based data assimilation at a later stage. Our motivation is that the data-assimilation will correct for the temporal differences in relative biomass production. We wanted to test if the model performs accurately with this generic crop, to see if we can continue with the data-assimilation. We realize that this motivation has not been mentioned clearly in our previous manuscript. We added the following text to the introduction:

L65-L74:

*"A confrontation between model simulations and satellite data to evaluate or update the model simulations is not always trivial. Most importantly, the magnitude of model simulations and satellite retrievals of soil moisture or biomass are often not directly comparable. Biases between models and observations are inevitable, because they represent different quantities (Koster et al., 2009, Reichle et al., 2004) or are simply based on different assumed parameterizations. Via parameter estimation, soil and vegetation parameters can be spatially tuned to reduce such biases, but this is often not feasible for satellite retrievals or difficult with more detailed models at the regional to global scale. For this same reason, state-of-the-art data assimilation systems for state updating are designed to correct for random error, and not for systematic bias. Therefore, satellite products of relative soil water indices or anomaly total water storages are often distributed (Wagner et al., 1998, Albergel et al., 2008, De Lannoy et al., 2016, Li et al., 2019), and the performance of large-scale model simulations is often evaluated using bias-free temporal skill metrics (De Lannoy et al., 2015, Gruber et al., 2020)."*

**L79: I don't understand the discretization of the soil column. It looks like no soil layers are used? But the root zone is divided into compartments? How are these discretized? Is the topsoil/subsoil information of HWSD (described later, L132) assigned to these compartments or are soil properties assumed to be homogeneous?**

Answer: In AquaCrop, output for soil moisture is given for the entire root-zone, but also for the different layers of the root zone, with compartments of 0.1m depth. The top layer (WC01) was used to evaluate against satellite data, which is equal to the top 10 cm of the soil. The text has been updated as follows:

L96-L100:
*"To calculate the soil water balance, AquaCrop divides the soil profile into multiple compartments (default 12) with depth increments Δz (default 0.1 m). For deeper soils, Δz increases exponentially with increasing soil depth, so that the processes of the near surface layers can still be resolved with sufficient detail. The number of compartments is independent of the number of soil horizons and the hydraulic properties for each compartment will be used depending on the soil layer in which they reside."*

L144-L145:

*"the AquaCrop volumetric moisture content of the top soil compartment (WC01), at 5cm depth (center of top 10 cm), was chosen for evaluation in this study."*

**L101: can you elaborate on what a field and a grid cell really are in AquaCrop. Most gridded approaches (or in fact also field-scale approaches) simulate one single point and assume it is representative for the field/grid cell. Is that similar here or is there any lateral heterogeneity**

considered within the simulation units (field or grid cell)? If not, this "replacement" merely affects how outputs are interpreted, no?

Answer: Indeed, each field is considered homogeneous as it would be in the original AquaCrop model. This imperfect mapping adds representativeness error and is mentioned as extra motivation to focus on skill metrics that do not include a bias component (R, anomR, ubRMSD). The following text has been included for clarification:

L86:

*"… for a single field, which is assumed to be homogeneous."*

**L116: this section should include the time period that is actually simulated. Or a modeling protocol section should be added that describes the simulation experiment(s) – simulations done in advance for "tuning", central simulation**

Answer: A new section has been added to the methods, describing the simulation setup.

L137-L151:

*"The model domain of this study covers the agricultural land in the central part of Europe (35°N-55°N, 10°E-20°E), and 45 pixels across all of mainland Europe where in situ soil moisture data are available for evaluation (three in situ points are also included in the central European domain). The model was run for the years 2011 through 2018, starting on the first of January 2011. The initial soil moisture content for the first year was set at $\vartheta\_FC$, since the runs were initiated mid-winter, and for the subsequent years the initial soil moisture content was based on the moisture content of the last day from the previous year. Because the evaluation for soil moisture was done with microwave-based satellite products that pertain to the surface layer, the AquaCrop volumetric moisture content of the top soil compartment (WC01), at a depth of 0.05 m (center of top 10 cm) was chosen for evaluation in this study. For the biomass, the daily productivity (t ha-1 ) was derived from the cumulative biomass. In the regional version of AquaCrop, a single homogeneous field is represented by a 30 arcsecond (~1-km) pixel, and input and output were defined independently for each pixel. The system can easily be set up for any given resolution over any domain. In this study, the model was run exclusively for dominantly rainfed agricultural areas, based on the land use map of the CORINE Land Cover inventory (Büttner, 2014) for the year 2012. This dataset is available at 100-m resolution and was aggregated to 30 arcseconds. To best represent the pixels as agricultural fields, only pixels were included of which at least 50 CORINE pixels (~50% of one AquaCrop pixel) contained non-irrigated agriculture."*

**L117: unclear why this input data set was selected and no bias correction was performed. See e.g. Ruane et al. 2021 (http://dx.doi.org/10.1016/j.agrformet.2020.108313) for the relevancy of input data sets. The AgMERRA data set (based on MERRA and available at 0.25° spatial resolution), would e.g. seem to be more suitable? Or other data sets described there or the ISIMIP3a input data sets that also cover 2012?**

Answer: There are many global datasets available for the meteorology. MERRA-2 is a well-established and in depth evaluated long term global re-analysis product, assimilating observations from satellite and gauge stations for precipitation, with a low product latency (Reichle et al. 2017 doi: https://doi.org/10.1175/JCLI-D-16-0720.1). AgMERRA is based on an older version of MERRA, using an outdated modelling system, artificially mapped to 0.25°, and only available until specific years (2010). MERRA-2 has a latency of at most a month, and if we wanted to, we could directly replace MERRA-2 re-analysis with real-time forecasts, which is beneficial for future simulations.

A comparison of MERRA-2 temperature and precipitation and data from field stations (in Austria and Czech Republic), showed satisfactory results. We are looking for reliable datasets at finer resolutions for future applications.

The text has been updated as follows:

*L168:*

*"The MERRA-2 meteorological variables have a 3-hourly temporal resolution, a spatial resolution of 0.5° lat x 0.625° lon, and are readily available at a latency of about a month."*

and in the discussion:

L403:

 *"The use of high resolution meteorological forcing is likely to be most important step to further improve fine-scale AquaCrop simulations."*

**L136: but some crops are grown throughout winter in Europe, how are these growing season treated?**

Answer: In the methods section (L228) we state that this 'generic crop' is suitable for the summer growing season. We therefore exclude analysis over the winter period (Nov-Feb). A different crop file would be needed to simulate the winter crops, also with a different starting date of the year (November).

**L140: is this some form of calibration? Can you better describe how the 30% were approximated? The GAEZ data would also provide some information on soil fertility that could help to abandon a guestimate.**

Answer: Yes, this was tuned (manually) by comparing for several locations the biomass production to the CGLS-DMP product. This value was also chosen as in AquaCrop, the recommended soil fertility is in the range of moderate-good instead of perfect (=0% reduction). The reduction of 30% falls within that recommended range.

We have updated this in the text as follows:

L187-L190:

*"The default of this parameter is 0%, referencing to unlimited soil fertility with the perfect concentrations of plant nutrients. Since this situation is very rare in real fields, even for well-maintained land, the value was manually tuned to 30% after initial model evaluation of daily biomass production with the CGLS-DMP (see section 3.1) product for several pixels."*

**L144: It's a bit funny to justify the choice of the generic crop with anticipated results. Also, it seems that this is speculation as no counterfactual experiment was conducted. Maybe more specific crop representation would substantially improve model performance? There clearly is still room for improvement in model performance, so we cannot say if this generic crop choice is a good one.**

Answer: We agree that the use of specific crops would most likely result in better model performances, however this is not within the scope of our research, and even not needed for future data assimilation (state updating) experiments, if designed to focus on relative temporal variabilities.

**L145: "this file was minimally tuned" this is unclear and needs a clearer description and justification. Keep in mind that the work should be reproducible.**

Answer: The minimally tuning refers to setting the start of the senescence stage, which was done after the comparison with CGLS-DMP. We have now specified this in the updated manuscript.

*L196-L197:*

*"After visual model evaluation and quantitative comparisons against satellite-based dry matter productivity (DMP, see below; Smets et al., 2019), the date of senescence was tuned manually, to optimally capture the length of the growing season."*

**L150: I don't understand why the canopy development had to be prescribed in calendar days. How is it normally done in AquaCrop? A typical approach to simulate phenological development is based on growing-degree-days (see e.g. Wang et al. 2017 http://dx.doi.org/10.1038/nplants.2017.102). Please add a description of how crop phenology is modeled and if/how that relates to transpiration.**

Answer: Indeed, growing degree days are the default option in AquaCrop. However, since we wanted to simulate the biomass production from the 1st to the last day of the year, we had to fix the length of the crop cycle on 365 days to avoid that in a cold year the length would exceed the 365 days of the year (and vice versa in a warm year). However, since in the calculation of crop transpiration growing degree days are considered, an over- or underestimation of the canopy cover will have only a small impact on the simulated crop transpiration (and hence also the effect on the simulated soil water content and biomass production will be limited).

*L201-L206:*

*"Because of this fixed annual cycle, the canopy development had to be simulated in calendar days instead of the more commonly used growing degree days. This results in an error in the simulation of canopy expansion during cold periods, but due to the consideration of growing degrees in the simulation of crop transpiration with the cold stress factor, the reduced biomass production in these periods is still captured. As can been seen from equations 1 and 2, the factors that affect the crop development, simulated by canopy cover, are soil water stress and cold temperature stress."*

**L151: the effect of GGD on transpiration is not clear from equation 2. How is this considered? Please elaborate to make sure that the model description in complete and understandable.**

Answer: Thank you for noticing. The cold stress coefficient is calculated with growing degree days, which largely eliminates the error of canopy cover in calendar days. This important variable was accidentally not included in equation 2, but that has now been corrected (L92).

**L156: what classes? And why 50 as a minimum? Are you saying "if at least half of the pixel is used for rainfed agriculture, it is included in the simulation set"? I wonder if that does not exclude a lot of cropland from the analysis? Is this necessary to have a clear signal from cropland when comparing to satellite data? Please expand.**

Answer: There would indeed be some crop areas lost. However, since a 1-km pixel should represent an agricultural field, for pixels in which more than 50% is covered by another type of vegetation or not even by vegetation but e.g. an urban class, this would no longer be a realistic representation of an agricultural field. This part has been rephrased and moved to the model setup description

*L149-L151:*

*"This dataset is available at 100-m resolution and was aggregated to 30 arcseconds. To best represent the pixels as agricultural fields, only pixels were included of which at least 50 CORINE pixels (~50% of one AquaCrop pixel) contained non-irrigated agriculture."*

**L187: What is the outcome of that implicit quality screening?**

Answer: This sentence has been removed to avoid confusion.

This statement was too trivial: since we are not simulating over areas where satellite soil moisture retrievals do not perform well (e.g. ice, urban, complex terrain), we are also not using any compromised satellite retrievals.

**L200: is there a name and reference for the "recommended conservative quality screening"? How important is that for the outcome?**

Answer: There is; O'Neill, Peggy, et al. "Algorithm Theoretical Basis Document. Level 2 & 3 Soil Moisture (Passive) Data Products." (2018). We have added this reference to the updated manuscript in L256. Microwave satellites are sensitive to various aspects (frozen soils, snow, steep slopes, urban area, water bodies). Quality control is done for both the CGLS-SSM and SMAP-SSM, to minimize trivial product errors.

**L214: this information should be moved and expanded to the methods section. Modeling domain, selection of grid cells included in simulations, temporal range covered etc. need to be described in the methods part.**

Answer: This has been updated accordingly in the revised manuscript, see above.

**L216: all metrics should be described with equations. To me, at least, it is unclear how anomalies were computed or how the bias was removed – it seems there are different options to do so.**

Answer: The following equations have been added to the revised manuscript:

L273-L278:

$$Bias = \frac{1}{N}\sum_{n=1}^{N}(x_n - y_n)$$

$$RMSD = \sqrt{\frac{1}{N}\sum_{n=1}^{N}(x_n - y_n)^2}$$

$$ubRMSD = \sqrt{RMSD^2 - bias^2}$$

$$R = \frac{\frac{1}{N}\sum_{n=1}^{N}(x_n - \bar{x})(y_n - \bar{y})}{\sqrt{(\sum_{n=1}^{N}(x_n - \bar{x})^2(\sum_{n=1}^{N}(y_n - \bar{y})^2}}$$

Where x are the simulated output data from AquaCrop and y are the observations from the satellite products. $\bar{x}$ and $\bar{y}$ are the time mean values.

To calculate anomalies, we have included Gruber et al., (2020) (https://doi.org/10.1016/j.rse.2020.111806).

L284-L285:

*"… To obtain anomalies as described by Gruber et al. (2020)."*

**L228: multi-year => how many years?**

Answer: This has been updated

L283-L284:

*"…A multi-year climatology (8 years for CGLS-DMP, 4 for CGLS-SSM and 3.5 for SMAP-SSM)"*

**L233: the median of the matching 10 days?**

Answer: the sentence has been rephrased:

L289:

*"…the median of the matching 10-day period"*

**L236: I don't understand the exact processing or the intentions of the correlation analysis of AEI and the satellite products**

Answer: The AquaCrop model is run without any irrigation applications. In areas where irrigation is very common, such as northern Italy, this could cause a mismatch between observations from the satellite and model simulations. Since it is impossible to know the exact dates and amounts of irrigation at this scale for each location, the Area Equipped for Irrigation (AEI) dataset was used, to distinguish regions of high irrigation potential. By comparing correlation coefficients to the percentage of AEI, we could say that there is a possible effect of irrigation that influences the correlation between the model and satellite retrieval products. The following text has been added for clarification:

L290-L292:

*"In this study, only rainfed agriculture is considered. However, it is very likely that irrigation will occasionally take place on rainfed fields, where the timing and volume is based on local decisions made by farmers. Irrigation practices were not included in the model simulations. To analyse how this human-driven process could influence the model performance,…"*

**L237: the metrics section fails to explain how 5cm soil moisture values are made comparable to simulated soil moisture of the rooting zone? Maybe this is not an issue and just a problem of me not understanding how the soil is discretized in AquaCrop (see comment above)**

Answer: Please see answer above, for L79.

**L257: where can we see this? How do you know?**

Answer: This could be visualized with maps presenting water stress and temperature stress over the domain. You would typically find higher water stress in the South, due to more drier periods, and cold temperature stress in the North.

We have added a typical timeseries for a location in the South to visualize the main difference between the North and South of Europe (Fig. 4).

**L258: "tons" are abbreviated with small-caps t (also in legend of Fig 1 and titles of Fig 2)**

Answer: the units have been made consistent.

**L267: what is "high rate of rainfall"? Do you refer to the precipitation intensity or total amounts? That part of Germany included in your modeling domain and Poland are not exactly areas of high rainfall.**

Answer: This sentence was not well phrased and has been updated in the revised manuscript. In areas where crop growth is expected as there is sufficient water supply by precipitation, this supply will be drained in extremely sandy soils, therefore still create water stress for the plant.

L330-L331:

*" In areas where there is a sufficient amount of rainfall, but soils are typically sandy and have a low TAW and high Ksat, soil water easily drains through the profile, which prevents optimal crop production."*

**L268: why should the satellite not see water stress effects? Please elaborate**

Answer: The DMP is not a direct satellite observation, but a product that uses fAPAR from optical satellites and variables from other resources (meteo data from ECMWF) to derive the productivity. It is true that the stress should be observed in FAPAR at some stage (it is a 10-daily product), but the DMP manual still emphasizes that this product contains several limitations, such as omitting water stress and nutrient deficiencies. This is better explained in the revised manuscript.

L214-L219:

*"The CGLS-DMP is based on a simplified Monteith (1972) approach that makes use of the fraction of absorbed photosynthetically active radiation (fAPAR), which is derived from the optical satellite missions Satellite Pour l'Observation de la Terre (SPOT; 1999-2014) and Project for On-Board Autonomy - Vegetation (PROBA-V; 2014-June 2020), ECMWF re-analysis estimates of atmospheric variables such as radiation and temperature, and land cover information from the ESA CCA Land Cover Map. The retrieval algorithm is thus driven by atmospheric water availabilities, without explicitly accounting for water storage in the soil."*

**L280: is this a finding by eyeball? Please provide metrics that support your claim (e.g. distribution of errors per texture class or similar).**

Answer: The particular erroneous output stands out in the maps, and corresponds exactly to a certain soil type, with 93% of sand. This is a unique soil class from the HWSD, which is not suitable to be used in AquaCrop. Please see answers to L330 for additional information.

**L285: My understanding of the methods section is that only pixels with at least 50% rainfed agriculture according to the CORINE data set are included in the simulations. How come you now claim that the sandy regions included in the analysis (and thus containing agricultural land) are not suitable for agriculture?**

Answer: This is something we did not expect to see either. However, it could be possible that the dominant soil class over an area is not the dominant soil class for the agricultural areas within that pixel. This soil class of 93% sand is simply an outlier in the soil classification, i.e. in our soil input data set. Using multiple sources of datasets with different grid projections could result in such mismatches, as is now mentioned in the discussion (L. 408-410).

**L292: just to avoid ambiguity: these R values refer to AquaCrop vs. in-situ and AquaCrop vs. CGLS-SSM, correct? Not to in-situ vs. CGLS-SSM in the second case?**

Answer: Figure 6.a. shows AquaCrop vs. in situ. 6b AquaCrop vs. CGLS-SSM & AquaCrop vs. SMAP-SSM, 6c. in situ vs. CGLS-SSM & In situ vs. SMAP-SSM. The figure titles have been changed to make it clearer.

**L294: why 42 pixels? In section 3.4 you describe 32 for SMAP-SSM and 42+3 for CGLS-SSM. The extra 3 HOL points are not available for SMAP-SSM?**

Answer: Indeed, there is no SMAP-SSM data over the HOAL locations, so we could not make a comparison there. The datapoints are now consistent.

**L295: is this the mean R of the temporal correlation averaged across the sites or a correlation across time and space?**

Answer: It is a temporal correlation averaged across the sites; we have fine-tuned the language in the revised manuscript (L365).

**L306: this seems to miss a reference? Or are these your own findings?**

Answer: A reference has been added (L377).

**L315: I don't understand this sentence. Irrigation could also dampen the amplitude making the overall weather-driven signal (the only aspect captured by AquaCrop) smaller compared to the noise and thus harder to get any correlation at all. Also for a comparison between the different data sets it seems difficult to compare across different samples? Why not make these claims based on the same spatial mask?**

Answer: Limiting the analysis to the same spatial mask would not be beneficial, because active and passive microwave-based soil moisture retrievals each have their own overpass times and limitations (recommended quality flags) in space and time. Crossmasking both datasets, possibly also with optical satellite-based biomass retrievals, would mean a too large loss of valuable data. We will add a note about this in the manuscript. The text has been rephrased:

L386-L388:

*"…even if the simulations were limited to dominantly rainfed agricultural areas according to the CORINE land use map and therefore did not include irrigation, it is possible that in reality irrigation is occasionally applied in rainfed fields and seen by the satellite data, resulting in lower correlation metrics."*

**L323: I did not see any analysis separating seasonal, interannual and short-term temporal dynamics here. What results support this claim here?**

Answer: With Pearson's R we compared the seasonal dynamics of the model and the product. The anomaly correlations account for interannual temporal dynamics, as the seasonally varying climatology has been subtracted for each year. The time series shown in the manuscript then indicate the short-term and interannual temporal dynamics. This is mentioned in L. 282.

**L330: Is this speculation or can this be shown somewhere? A simple test could be to run the model with uniform soil parameters – the computational costs are claimed to be low?**

Answer: The TAW map is presented in Fig. 3a as a comparison. We see a clear agreement with the lower anomR biomass map in the North (North Germany/ Poland) and low TAW values and we verified this in the case for extremely sandy soil class, which contained very low TAW values (due to low FC). As an extra confirmation, we grouped the anomaly correlations for the Northern half of the study domain (>46°N) by TAW values which shows that lower anomaly correlations are generally found in TAW values up to 0.1 $m^3 m^{-3}$ (see Fig. 1). This figure is not included in the revised manuscript.

[Figure]

***Figure 1*** *Boxplots of anomaly correlations for Biomass productivity grouped by ranges of TAW over northern half of the study domain (46°N – 55°N).*

**L338: again here seems to be a reference missing? The comparison of the skill of the AquaCrop model to the 2 different data products should be conducted on the same spatial mask.**

Answer: A mask would indeed be better to confirm that statement. However, with the intercomparison of the in situ products, we already make a comparison over the same locations for both SMAP-SSM and CLGS-SSM, showing much better correlations of SMAP-SSM to both in situ observations and AquaCrop simulations. We kept the higher resolution maps for CGLS-SSM as visualization in the manuscript, but we now also aggregated the CGLS-SSM to the EASE2 grid and the same spatial mask is used as for SMAP-SSM. The resulting mean correlations are now added to the manuscript.

L340-L343:

*"When aggregating CGLS-SSM to the EASE2 9-km grid using the same spatial mask of SMAP-SSM, temporal correlations with AquaCrop increase slightly, with a mean R of 0.57 (spatial standard deviation STDEV: 0.08) and a mean anomR of 0.56 (STDEV: 0.06) (not shown), and remain well below the correlation values between SMAP-SSM and AquaCrop SSM."*

The correlation maps of CGLS-SSM on the EASE2 grid are shown below (not added to the manuscript).

[Figure]

**Figure 2** CGLS-SSM aggregated to EASE2 grid of CGLS-SMAP, on the same spatial mask. Shown are the temporal correlations with aggregated AC SSM.

**L341: This point was also made earlier on, but I still don't understand how the explicit focus on "agriculture-only" pixels can include substantial areas with soil parameters unfit for agricultural production?**

Answer: Please, see comment above at L285. In other words: parameter data sets are by design not self-consistent, which has now been mentioned in the discussion (L408-410).

**L342: Speculation or finding? Is that a scientific debate or finding that could be referenced or would e.g. the SMAP-SSM quality flags suggest such relationship?**

Answer: The sentence has been rephrased.

L451-L452:

*"At the same time, satellite-based soil moisture retrievals also contain errors related to a priori defined soil hydraulic parameters."*

**L345: There are already data on varying crop parameters such as growing seasons and fertilizer available and I guess the abandonment of the idea of a "generic crop" would be a prerequisite to introduce time- and space-variant crop parameters.**

Answer: Agreed, however due to reasons mentioned earlier we prefer to stay with the 'generic crop'.

**L348: this is an interesting idea, but can you elaborate on this a bit to make it more tangible?**

Answer: We have made changes to the manuscript to better clarify our intentions for the satellite-based data assimilation, in the new discussion section 4.4.

**Fig 2: Panel b shows productivity not production? Small-caps t not T for tones.**

Answer: has been changed.

**Fig 3: What are white areas? In figs 1+2 I was assuming that these are pixels not simulated (could be explained in methods section), but is the SMAP-SSM data set more patchy or are these excluded for some other reason? This should be explained in the caption.**

Answer: The white areas are indeed no data areas. We have specified this in the revised figure descriptions. Due to the strict quality screening for SMAP-SSM, there is less data available.

**Fig 4: panels b and c don't show information on agreement across the 3 product classes (satellites, AquaCrop, in situ) or if there is always just 2 of the 3 agreeing with each other. That would have implications for the interpretation of results, wouldn't it?**

Answer: See above. With 4.b (now 6.b). and 4.c (now 6.c) we wanted to visualize that SMAP-SSM in most cases correlates better to in situ points and to the AquaCrop measurements than the CGLS-SSM product does.

**Reviewer #2 (Anonymous):**

The authors would like to thank the reviewer for their comments and recommendations to improve our manuscript. We have considered each comment in the revised manuscript and would like to provide an overview of the adjustments.

**The manuscript by Shannon de Roos et al. documented an effort on assessing the performance of AquaCrop model simulation at regional scale with benchmark of remotely sensed and in-situ observations of biomass and soil moisture. Generally the current manuscript is lacking scientific interpretation and insights. I have the following concerns for the authors to consider:**

**First, I suggest the author further consolidate their objectives of their manuscript. It reads like that the authors are going to address the scaling issue from point to global models in the introduction part, but the results only stay at model evaluation at a fixed scale (i.e. 1km). If the objective is scaling, the claim that "the regional AquaCrop model proves to be useful in assessing crop production and soil moisture at various scales and could serve as a bridge between point-based and global models" is not well backed up by the analysis in the manuscript. There is only one scale for model simulation in the current manuscript, i.e. 1km scale. Most importantly, it is unclear how the regional model simulation can serve as a bridge between point-based and global models. The scaling issue from point-based to global models is not touched in this manuscript at all, but deserves a further investigation in the framework the authors developed. For example, when assessing the soil moisture, the authors aggregated the 1km soil moisture simulation to 9km. In other model setups like GGCMI, the model would be running at 9km or even larger scales. How do the performances of different model setups vary and what are the controlling factors for those performance variations? I think those are the key questions to be answered and would also be more interesting for the crop model and land surface modeling communities. The authors can actually test those questions for both biomass and soil moisture simulations with their regional model simulation platform.**

**If the objective is model evaluation, I suggest the authors rewrite their motivation part in the introduction. Model evaluation with newly remote sensing data is also interesting, especially in the context of further data assimilation (as indicated by the authors in the conclusion part) experiments in which we need to have some information about the model uncertainties.**

Answer: We agree that the introduction needed a better clarification and have rewritten the introduction and added the following text:

L39-L46:

*"A possibility to correct for scaling errors, is the updating of the model simulations with remote sensing observations via data assimilation. There are several studies that have already used data assimilation in regional crop modelling systems (De wit & van Diepen, 2007; Mladenova et al., 2019; Zhuo et al., 2019), either for parameter or state updating. Parameter updating or calibration allows to match the absolute values of the simulations with (most often historical) observations. State updating allows to correct the relative temporal evolution and to obtain better initial conditions for subsequent model predictions. To get the most optimal results with data assimilation, it is important to start with a reliable model that is able capture the seasonal as well as interannual temporal variabilities."*

*"To evaluate, or later update, select variables within such a regional modeling system, in situ data only provide sparse information. However, a range of optical and microwave-based satellite data are available at various temporal and spatial resolutions. A confrontation between model simulations and satellite data to evaluate or update the model simulations is not always trivial. Most importantly, the magnitude of model simulations and satellite retrievals of soil moisture or biomass are often not directly comparable. Biases between models and observations are inevitable, because they represent different quantities (Koster et al., 2009, Reichle et al., 2004) or are simply based on different assumed parameterizations. Via parameter estimation, soil and vegetation parameters can be spatially tuned to reduce such biases, but this is often not feasible for satellite retrievals or difficult with more detailed models at the regional to global scale. For this same reason, state-of-the-art data assimilation systems for state updating are designed to correct for random error, and not for systematic bias. Therefore, satellite products of relative soil water indices or anomaly total water storages are often distributed (Wagner et al., 1998, Albergel et al., 2008, De Lannoy et al., 2016, Li et al., 2019), and the performance of large-scale model simulations is often evaluated using bias-free temporal skill metrics (De Lannoy et al., 2015, Gruber et al., 2020)."*

**Third, it seems that the transpiration simulation in AquaCrop plays a very important role in simulating biomass and soil moisture. Why not do some assessment on transpiration simulation with flux tower and remote sensing ET data?**

Answer: Thank you for the suggestion, such an analysis could indeed be of extra value. However, the current options to access evaporation and transpiration data are not ideal. We could make use of the datasets from FLUX towers over Europe (at cropland sites), but the data accessibility and availability is somewhat limited for our simulation period. Additionally, these data would only enable to evaluate the total evapotranspiration, not transpiration separately. As an alternative, the evapotranspiration data from GLEAM are satellite- and model-based, and offer separate transpiration and evaporation estimates for our study domain and period. However, these data are produced at a much coarser resolution (25 km) than our model simulations. For the reasons previously mentioned, we would again resort to evaluation metrics such as correlation and ubRMSD. In short, if necessary we will include an evaluation of (evapo)-transpiration, but we believe that it might inevitably be less comprehensive than the evaluation we already provided with a range of soil moisture data. Therefore we have included the following text in the discussion:

*"The evaluation was limited to surface soil moisture and biomass, but could be further expanded to other variables such as root-zone soil moisture and transpiration in the future. Reference data for the latter variable are always informed by strong (often modeled) background information (Martens et al., 2017 , Reichle et al., 2019) and not directly observed over large regions."*

**Forth, the authors directly jumped to conclusion after showing their results. Are there any insights to be discussed from this model evaluation effort? I suggest the authors bring up their most important findings and give more implications about crop model set up and evaluation at regional**

**scale in the discussion part, which is now totally missing. Otherwise, the scientific merit of this manuscript is largely limited.**

Answer: Thank you for pointing this out. In the revised version, we have added a discussion section (section 4.3) of the model findings in relation to this regional model set-up, dissecting the advantages and possible improvements:

L390-L425:

*"4.4 Discussion of the regional AquaCrop model*
*The current gridded AquaCrop model has several conveniences, such as the efficient parallel processing structure, the ability to run at any resolution and domain, and the modular setup in which a compiled executable can be easily replaced by newer AquaCrop versions. The model setup is chosen to facilitate subsequent embedding within a future satellite-based data assimilation system.*
*The regional modelling system was designed to capture the seasonal and inter-annual variability, with some important simplifications. A general C3 crop was assumed and management data was considered as homogeneous over the entire study area, whereas meteorology and soil information were spatially variant. Therefore, the evaluation of this regional crop model setup against satellite products was mainly done in terms of unbiased temporal metrics. AquaCrop accurately simulates the temporal variability in biomass and surface soil moisture, especially in the northern regions and if the soil's TAW is not limiting. Limitations in the accuracy of the input precipitation (MERRA-2) causes slightly worse simulations in the water limited southern regions, where biomass shows a fast response to limited (but sometimes inaccurately timed) rainfall events. The use of high resolution meteorological forcing is likely to be most important next step to further improve fine-scale AquaCrop simulations. The evaluation was limited to surface soil moisture and biomass, but could be further expanded to other variables such as root-zone soil moisture and transpiration in the future. Reference data for the latter variable are always informed by strong (often modelled) background information (Martens et al., 2017 , Reichle et al., 2019) and not directly observed over large regions.*
*The suitability of this model system to estimate the spatial variability in soil moisture and yield production for specific crop types would require further analysis and more detailed input information. For example, by combining input datasets from different sources, some unsuitable cropland areas were identified (e.g. too low TAW in combination with high Ksat) that were not filtered out from this analysis. Furthermore, unmodeled irrigation could influence the regional model performance. Most importantly, the relative spatial variability in biomass is likely not dominated by meteorology and soil texture, but by the various types of crops. The parameters associated with each of these crops could be spatially optimized (calibration, data assimilation for parameter estimation) in future work, using historical time series of spatially covering reference data, e.g. optical Sentinel-2 data.*
*The regional model evaluation could only be performed with satellite retrievals, but such an evaluation is limited to the days of overpass, and to times and locations where retrievals are of sufficient quality. For example, SMAP-SSM retrievals are filtered out under too dense vegetation or frozen conditions. Furthermore, the satellite signal may represent a slightly different quantity than what is modelled. Additionally, microwave signals only pertain to the upper 5 cm of the soil, but the model's surface layer is 10 cm. The provided quality flags on CGLS-SSM are less strict, providing a better spatial coverage of fine-scale data. However, the C-band soil moisture measurements pertain to an even shallower soil depth and are likely more affected by vegetation. In any case, both the satellite retrievals and model simulations have their own systematic and random errors. The influence of the former is suppressed in*

*this study by focusing on relative temporal variability. To further dynamically improve model simulations, or to add value to the available satellite data (e.g. dynamically interpolate) via AquaCrop modelling, random errors in both sources can be limited via data assimilation for state updating."*

Other comments:

**L80-81: please specify the soil layer depths you used in your regional setup. This is critical information when you want to compare your simulation with satellite-based soil moisture retrievals.**

Answer: clarifications about the soil layer depth have been included in the revised manuscript:

L144-L145:

*"the AquaCrop volumetric moisture content of the top soil compartment (WC01), at a depth of 0.05 m (center of top 10 cm) was chosen for evaluation in this study."*

**L81-L85: more description about hydrology (runoff, percolation, …) in the model is required as evaluating soil moisture simulation performance is an important component of this manuscript.**

Answer: A description of each component in the water balance has been added to the revised manuscript.

L100-L107:

*"The simulation of the water content in each compartment is done with a set of finite difference equations (subroutines), that are defined in terms of the dependent variable $\vartheta$, as represented in equation 3 (Raes, et al., 2012). First, the drainage of the soil profile is calculated. Then, the water infiltration is computed (after subtraction of surface runoff) and upward movement of water by capillary rise is estimated. Finally, the amount of water lost by evaporation and crop transpiration is subtracted.*

$$\theta_{i,j} = \theta_{i,j-1} + \Delta\theta_{DF_{i,\Delta t}} + \Delta\theta_{I_{i,\Delta t}} + \Delta\theta_{CR_{i,\Delta t}} + \Delta\theta_{E_{i,\Delta t}} + \Delta\theta_{T_{i,\Delta t}} \qquad (3)$$

*where $\theta_{i,j}$ is the soil water content of compartment i at time step j, $\theta_{i,j-1}$ is the water content of compartment i at the previous time step and $\Delta\theta_{X_{i,\Delta t}}$ indicate the change in moisture due to various processes X, with X=DF: downward flow, I: infiltration, CR: capillary rise, E: soil evaporation, T: crop transpiration."*

L114-L123:

*"Infiltration (I) is the sum of water that enters the soil, which is rainfall minus surface runoff, and possibly irrigation. Internal drainage between compartments is defined by the drainage ability, which depends on $\theta_{FC}$ and $\theta_{sat}$ (eq. 4). The cumulative drainage amount from any compartment will percolate through as long as its drainage ability is greater than or equal to the drainage ability of the overlying compartment. If the drainage ability is lower than the overlying compartment, the cumulative drainage amount will be stored in that compartment, increasing the water content and thereby its drainage ability. If then the drainage ability is reaching the equal amount of that of the overlying compartment, excess drainage will percolate through to the lower compartment. For the bottom soil compartment, the drainage is lost as deep percolation. The runoff is estimated based on the curve number (CN) method, developed by the US Soil Conservation Service (USDA, 1964). The CN values are dependent on*

*the saturated hydraulic conductivity of the topsoil layer. Upward flow by capillary rise is estimated based on the depth of the groundwater table and hydraulic characteristics of the soil layers."*

**Section 2.2: it would be good to have a flowchart for the regional setup.**

Answer: Thank you for the suggestion, a flowchart (Fig. 1) has been added to the revised manuscript.

[Figure]

***Figure 3*** *Flowchart of regional AquaCrop model setup.*

**L277-L278: how about also aggregating CGLS-SSM to 9km and compare it with model simulations at the same scale with SMAP data? That would be a more fair comparison.**

Answer: Thank you for the suggestion. An extra analysis has been made in which aggregated CGLS-SSM has been compared to AquaCrop SSM on the same EASE2 Grid as SMAP-SSM. This results of this analysis are now mentioned in the revised manuscript.

L340-L343:

*"When aggregating CGLS-SSM to the EASE2 9-km grid using the same spatial mask of SMAP-SSM, temporal correlations with AquaCrop increase slightly, with a mean R of 0.57 (spatial standard deviation STDEV: 0.08) and a mean anomR of 0.56 (STDEV: 0.06) (not shown), and remain well below the correlation values between SMAP-SSM and AquaCrop SSM."*

The correlation maps of the aggregated CGLS-SSM product are shown below, but are not added to the revised manuscript.

[Figure]

**Figure 4** CGLS-SSM aggregated to EASE2 grid of CGLS-SMAP, on the same spatial mask. Shown are the temporal correlations with aggregated AC SSM.

---

## Author Response (AR2)

The authors would like to thank the reviewer and editor again, for their quick reply on our revised manuscript submission.

**Please address the following comments from a second review by one of the original reviewers. I agree with the reviewer's comment about extra figures in response letters. Please include these figures in either the main text or in supplementary material.**

Answer: The two extra figures have now been added to the manuscript (Fig. 4 and Fig. 7).

**Many of my original concerns have been addressed adequately by the authors, but a few points remain.**
**A) I'm still not convinced by the generic crop approach. I understand the reasoning given in the answer (page 4 of the response letter), but the modifications of the text do not even mention the choice of a generic crop in the model setup. Also, I do see a mismatch between the evaluation (performance analysis) done here (which suffers from mismatches in crop parameterization) and a – hypothetically anticipated – data assimilation (where such mismatches may be eliminated by parameter updating or state updating). So how is this addressed here? I see no clear justification of this choice in the introduction/methods nor a discussion of the implications later on.**

Answer: Clarifications have been added in the abstract, methods and discussion, to emphasize our choice for the generic crop.

L8-9

"The setup with a generic crop is chosen as a baseline for a future satellite-based data assimilation system."

L73:

"The assumption of a generic crop will for example lead to inevitable biases."

L196-197

"Spatial and temporal gaps of information at the ~1-km resolution prevent the inclusion of a more detailed crop parameterization. Furthermore, this research is focused on capturing relative temporal variation in biomass (not yield) for future use in a data assimilation system, a generic crop was developed and used for the entire domain. It is expected that regional differences of crop productivity from different crops will be corrected for via future data assimilation."

L414-415

"Furthermore, applying crop specific parameters to the crop file would most likely result in better biomass and yield simulations, which would mainly improve the temporal bias and spatial performance metrics."

**B) The tuning to 30% (L187-190) needs to be either clearly declared as value arbitrarily chosen or a reference for the recommended range that is hinted at needs to be provided (and then the 30% need to be deduced from that).**

Answer: This value was chosen in discussion with AquaCrop developer Dirk Raes after manual calibration. It is therefore not completely arbitrary but supported by expert knowledge. The explanation in the text has been changed to:

L191-192

"…, which is a setting recommended by expert knowledge of the AquaCrop source code developers

**C) I generally disagree with the strategy to provide extra figures in response letters only. Assuming that other readers may have similar questions, they should be presented the same evidence not only the reviewer who is able to ask for it. If you deem these figures too unimportant to be included in the main text, please supply them in a supplement (or similar).**

Answer: The two extra figures have now been added to the manuscript (Fig. 4 and Fig. 7).